# Improving neural network representations using human similarity judgments

**Lukas Muttenthaler**[*]
Google DeepMind
Machine Learning Group
Technische Universität Berlin
BIFOLD[†]
Berlin, Germany

**Lorenz Linhardt**
Machine Learning Group
Technische Universität Berlin
BIFOLD[†]
Berlin, Germany

**Jonas Dippel**
Machine Learning Group
Technische Universität Berlin
BIFOLD[†]
Berlin, Germany

**Robert A. Vandermeulen**
Machine Learning Group
Technische Universität Berlin
BIFOLD[†]
Berlin, Germany

**Katherine Hermann**
Google DeepMind
Mountain View, CA, USA

**Andrew K. Lampinen**
Google DeepMind
London, UK

**Simon Kornblith**
Google DeepMind
Toronto, Canada

## Abstract

Deep neural networks have reached human-level performance on many computer vision tasks. However, the objectives used to train these networks enforce only that similar images are embedded at similar locations in the representation space, and do not directly constrain the global structure of the resulting space. Here, we explore the impact of supervising this global structure by linearly aligning it with human similarity judgments. We find that a naive approach leads to large changes in local representational structure that harm downstream performance. Thus, we propose a novel method that aligns the global structure of representations while preserving their local structure. This global-local transform considerably improves accuracy across a variety of few-shot learning and anomaly detection tasks. Our results indicate that human visual representations are globally organized in a way that facilitates learning from few examples, and incorporating this global structure into neural network representations improves performance on downstream tasks.

## 1 Introduction

Representation learning usually involves pretraining a network on a large, diverse dataset using one of a handful of different types of supervision. In computer vision, early successes came from training networks to predict class labels [47, 85, 28]. More recent work has demonstrated the power of contrastive representation learning [9, 29, 70, 8]. Self-supervised contrastive models learn a space where images lie close to other augmentations of the same image [9], whereas image/text contrastive models learn a space where images lie close to embeddings of corresponding text [70, 39, 68].

---

[*]Work done as part of the Google Research Collabs Programme.
[†]Berlin Institute for the Foundations of Learning and Data, Berlin, Germany.

37th Conference on Neural Information Processing Systems (NeurIPS 2023).

Although these representation learning strategies are effective, their pretraining objectives do not directly constrain the global structure of the learned representations. To classify ImageNet images into 1000 classes, networks' penultimate layers must learn representations that allow examples of a given class to be linearly separated from representations of other classes, but the classes themselves could be arranged in any fashion. The objective itself does not directly encourage images of tabby cats to be closer to images of other breeds of cats than to images of raccoons. Similarly, contrastive objectives force related embeddings to be close, and unrelated embeddings to be far, but if a cluster of data points is sufficiently far from all other data points, then the location of this cluster in the embedding space has minimal impact on the value of the loss [62, 35, 9].

Even without explicit global supervision, however, networks implicitly learn to organize high-level concepts somewhat coherently. For example, ImageNet models' representations roughly cluster according to superclasses [19], and the similarity structure of neural network representation spaces is non-trivially similar to similarity structures inferred from brain data or human judgments [40, 92, 66, 80, 49]. The structure that is learned likely reflects a combination of visual similarity between images from related classes and networks' inductive biases. However, there is little reason to believe that this implicitly-learned global structure should be optimal.

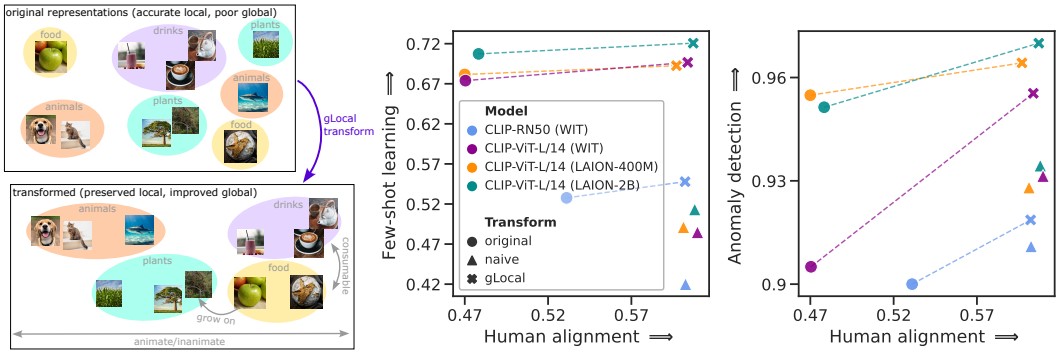

(a) Conceptual cartoon.  (b) Downstream task performance vs. human alignment.

Figure 1: Global-local (gLocal) transforms yield a *best-of-both-worlds* representation space, which improves overall performance. (a) The original representations capture local structure, such as that different trees are similar, but have poor global structure. The gLocal transform preserves local structure, while integrating global information from human knowledge; e.g., unifying superordinate categories, organizing by "animacy", or connecting semantically-related categories like "food" and "drink". (b) The gLocal transforms improve both human alignment and downstream task performance compared to original and naively aligned representations for image/text models. We report mean accuracies on anomaly detection and (5-shot) few-shot learning tasks.

Human and machine vision differ in many ways. Some differences relate to sensitivity to distortions and out-of-distribution generalization [21, 22, 25]. For example, ImageNet models are more biased by texture than humans in their decision process [22, 34]. However, interventions that reduce neural networks' sensitivity to image distortions are not enough to make their representations as robust and transferable as those of humans [23]. Experiments inferring humans' object representations from similarity judgments suggest that humans use a complex combination of semantics, texture, shape, and color-related concepts for performing similarity judgments [31, 60].

Various strategies have been proposed to improve *representational alignment* between neural nets and humans [e.g., 66, 67, 2, 61], to improve robustness against image distortions [e.g., 34, 21], or for obtaining models that make more human-like errors [e.g., 22, 26]. Muttenthaler et al. [61] previously found that a linear transformation learned to maximize alignment on one dataset of human similarity judgments generalized to different datasets of similarity judgments. However, it remains unclear whether better representational alignment can improve networks' generalization on vision tasks.

Here, we study the impact of aligning representations' global structure with human similarity judgments on downstream tasks. These similarity judgments, collected by asking subjects to choose the *odd-one-out* in a triplet of images [32], provide explicit supervision for relationships among disparate concepts. Although the number of images we use for alignment is three to six orders of magnitude smaller than the number of images in pretraining, we observe significant improvements. Specifically, our contributions are as follows:

- We introduce the *gLocal transform*, a linear transformation that minimizes a combination of a *global alignment loss*, which aligns the representation with human similarity judgments, and a *local contrastive loss* that maintains the local structure of the original representation space.
- We show that the gLocal transform preserves the local structure of the original space but captures the same global structure as a *naive transform* that minimizes only the global alignment loss.
- We demonstrate that the gLocal transform substantially increases performance on a variety of few-shot learning and anomaly detection tasks. By contrast, the naive transform impairs performance.
- We compare the human alignment of gLocal and naively transformed representations on four human similarity judgment datasets, finding that the gLocal transform yields only marginally worse alignment than the naive transform.

## 2 Related work

How can we build models which learn representations that support performance on variable downstream tasks? This question has been a core theme of computer vision research [9, 27, 58], but the impact of pretraining on feature representations is complex, and better performance does not necessarily yield more transferable representations [e.g., 44, 45]. For example, some pretraining leads to shortcut learning [23, 50, 1, 57, 22, 34, 3, 91]. Because the relationship between training methods and representations is complicated, it is useful to study how datasets and training shape representations [33]. Standard training objectives do not explicitly constrain the global structure of representations; nevertheless, these objectives yield representations that capture some aspects of the higher-order category structure [e.g., 40] and neural unit activity [e.g., 92, 80] of human and animal representations of the same images. Some models progressively differentiate hierarchical structure over the course of learning [5] in a similar way to how humans learn semantic features [73, 18, 78, 5, 79]. Even so, learned representations still fail to capture important aspects of the structure that humans learn [6]. Human representations capture both perceptual and semantic features [7, 69], including many levels of semantic hierarchy (e.g. higher-level: "animate" [11], superordinate: "mammal", basic: "dog", subordinate: "daschund"), with a bias towards the basic level [74, 55, 38], as well as cross-cutting semantic features [84, 56].

While models may implicitly learn to represent some of this structure, these implicit representations may have shortcomings. For example, Huh et al. [37] suggest that ImageNet models capture only higher-level categories where the sub-categories are visually similar. Similarly, Peterson et al. [66] show that model representations do not natively fully capture the structure of human similarity judgments, though they can be transformed to align better [66, 2]. In some cases, language provides more accurate estimates of human similarity judgments than image representations [54], and image-text models can have more human-aligned representations [61]. How does human alignment affect downstream performance? Sucholutsky & Griffiths [87] show that models which are more human-aligned (but not specifically optimized for alignment) are more robust on few-shot learning tasks. Other work shows benefits of incorporating higher-level category structure [51, 83]. Here, we ask whether transforming model representations to align with human knowledge can improve transfer.

## 3 Methods

**Data.** For measuring the degree of alignment between human and neural network similarity spaces, we use the THINGS dataset, which is a large behavioral dataset of $4.70$ million unique triplet responses crowdsourced from $12{,}340$ human participants for $1854$ natural object images [32]. Images used for collecting human responses in the triplet odd-one-out task were taken from the THINGS object concept and image database [30], which is a collection of natural object images.

**Odd-one-out accuracy.** The triplet odd-one-out task is a commonly used task in the cognitive sciences to measure human notions of object similarity without biasing a participant into a specific direction [20, 72, 31, 60]. To measure the degree of alignment between human and neural network similarity judgments in the THINGS triplet task, we examine the extent to which the odd-one-out can be identified directly from the similarities between images in models' representation spaces. Given representations $x_1$, $x_2$, and $x_3$ of the three images in a triplet, we first construct a similarity matrix $S \in \mathbb{R}^{3 \times 3}$ where $S_{i,j} := x_i^\top x_j / (\|x_i\|_2 \|x_j\|_2)$, the cosine similarity between a pair of representations.[3] We identify the closest pair of images in the triplet as $\arg\max_{i,j>i} S_{i,j}$ with the

---

[3]We use cosine similarity rather than the dot product because Muttenthaler et al. [61] have shown that it nearly always yields similar or better zero-shot odd-one-out accuracies.

remaining image being the odd-one-out. We define odd-one-out accuracy as the proportion of triplets where the odd-one-out matches the human odd-one-out choice.

**Alignment loss.** Given an image similarity matrix $\boldsymbol{S}$ and a triplet $\{i, j, k\}$ (here, images are indexed by natural numbers), the likelihood of a particular pair, $\{a, b\} \subset \{i, j, k\}$, being most similar, and hence the remaining image being the odd-one-out, is modeled by the softmax of the object similarities,

$$p(\{a,b\}|\{i,j,k\}, \boldsymbol{S}) := \exp(S_{a,b}) / \left(\exp(S_{i,j}) + \exp(S_{i,k}) + \exp(S_{j,k})\right). \tag{1}$$

For $n$ triplet responses we use the following negative log-likelihood, precisely defined in [60],

$$\mathcal{L}_{\text{global}}(\boldsymbol{S}) := -\frac{1}{n} \sum_{s=1}^{n} \log \underbrace{p\left(\{a_s, b_s\}|\{i_s, j_s, k_s\}, \boldsymbol{S}\right)}_{\text{odd-one-out prediction}}. \tag{2}$$

Since the triplets in [32] consist of randomly selected images, the concepts that humans use for their similarity judgments in the THINGS triplet odd-one-out task primarily reflect superordinate categories rather than fine-grained object features [31, 60], the above alignment loss can be viewed as a loss function whose objective is to transform the representations into a globally-restructured human similarity space where superordinate categories are emphasized over subordinate categories.

**Naive transform.** We first investigate a linear transformation that naively maximizes alignment between neural network representations and human similarity judgments with $L_2$ regularization. This transformation consists of a square matrix $\boldsymbol{W}$ obtained as the solution to

$$\arg\min_{\boldsymbol{W}, \boldsymbol{b}} \mathcal{L}_{\text{global}}(\boldsymbol{S}) + \lambda ||\boldsymbol{W}||_{\text{F}}^2, \tag{3}$$

where $S_{ij} = (\boldsymbol{W}\boldsymbol{x}_i + \boldsymbol{b})^\top (\boldsymbol{W}\boldsymbol{x}_j + \boldsymbol{b})$. We call this transformation the *naive transform* because the regularization term helps prevent overfitting to the training set, but does not encourage the transformed representation space to resemble the original space. Muttenthaler et al. [61] previously investigated a similar transformation. We determine $\lambda$ via grid-search using $k$-fold cross-validation (CV). To obtain a minimally biased estimate of the odd-one-out accuracy of the transform, we partition the 1854 objects in THINGS into two disjoint sets, following the procedure of Muttenthaler et al. [61].

**Global transform.** The naive transform does not preserve representational structure that is irrelevant to the odd-one-out task. The global transform instead shrinks $\boldsymbol{W}$ toward a scaled identity matrix by penalizing $\min_\alpha \|\boldsymbol{W} - \alpha I\|_{\text{F}}^2$, thus regularizing the transformed representation toward the original. The global transform solves the following minimization problem,

$$\arg\min_{\boldsymbol{W}, \boldsymbol{b}} \mathcal{L}_{\text{global}}(\boldsymbol{S}) + \lambda \min_\alpha \|\boldsymbol{W} - \alpha I\|_{\text{F}}^2 = \arg\min_{\boldsymbol{W}, \boldsymbol{b}} \mathcal{L}_{\text{global}}(\boldsymbol{S}) + \lambda \left\|\boldsymbol{W} - \left(\sum_{j=1}^{p} \boldsymbol{W}_{jj}/p\right) I\right\|_{\text{F}}^2. \tag{4}$$

We derive the above equality in Appx. F. Again, we select $\lambda$ via grid-search with $k$-fold CV.

**gLocal transform.** In preliminary experiments, we observed a trade-off between alignment and the transferability of a neural network's human-aligned representation space to downstream tasks. Maximizing alignment appears to slightly worsen downstream task performance, whereas using a large value of $\lambda$ in Eq. 4 leads to a representation that closely resembles the original (since $\lim_{\lambda \to \infty} \boldsymbol{W} = \sigma I$). Thus, we add an additional regularization term to the objective with the goal of preserving the local structure of the network's original representation space. This loss term can be seen as an additional constraint on the transformation matrix $\boldsymbol{W}$.

We call this loss term *local loss* and the transform that optimizes this full objective the *gLocal transform*, where global and local representations structures are jointly optimized. For this loss function, we embed all images of the ImageNet train and validation sets [17] in a neural network's $p$-dimensional penultimate layer space or image encoder space of image/text models. Let $\boldsymbol{Y} \in \mathbb{R}^{m \times p}$ be a neural network's feature matrix for all $m$ images in the ImageNet train set. Let $\boldsymbol{S}^*$ be the cosine similarity matrix using untransformed representations where $S_{ij}^* = \left(\boldsymbol{y}_i^\top \boldsymbol{y}_j\right) / \left(||\boldsymbol{y}_i||_2 ||\boldsymbol{y}_j||_2\right)$ and let $\boldsymbol{S}^\dagger$ be the cosine similarity matrix of the transformed representations where

$$S_{ij}^\dagger = \left((\boldsymbol{W}\boldsymbol{y}_i + \boldsymbol{b})^\top (\boldsymbol{W}\boldsymbol{y}_j + \boldsymbol{b})\right) / \left(||\boldsymbol{W}\boldsymbol{y}_i + \boldsymbol{b}||_2 ||\boldsymbol{W}\boldsymbol{y}_j + \boldsymbol{b}||_2\right).$$

Let $\sigma$ be a softmax function that transforms a similarity matrix into a probability distribution,

$$\sigma(\boldsymbol{S}, \tau)_{ij} := \frac{\exp(\boldsymbol{S}_{ij}/\tau)}{\sum_k^m \mathbb{1}_{[k \neq j]} \exp(\boldsymbol{S}_{ik}/\tau)},$$

where $\tau$ is a temperature and $\sigma(\boldsymbol{S}, \tau)_{ij} \in (0, 1)$. We can then define the local loss as the following contrastive objective between untransformed and transformed neural network similarity spaces,

$$\mathcal{L}_{\text{local}}(\boldsymbol{W}, \boldsymbol{b}, \tau) \coloneqq -\frac{1}{m^2 - m} \sum_i^m \sum_j^m \mathbb{1}_{[i \neq j]} \sigma(\boldsymbol{S}^*, \tau)_{ij} \log \left[ \sigma(\boldsymbol{S}^\dagger, \tau)_{ij} \right]. \qquad (5)$$

To avoid distributions that excessively emphasize the self-similarity of objects for small $\tau$, we exclude elements on the diagonal of the similarity matrices. The final *gLocal transform* is then,

$$\arg\min_{\boldsymbol{W}, \boldsymbol{b}} \quad \underbrace{(1 - \alpha) \, \mathcal{L}_{\text{global}}(\boldsymbol{W}, \boldsymbol{b})}_{\text{alignment}} + \underbrace{\alpha \mathcal{L}_{\text{local}}(\boldsymbol{W}, \boldsymbol{b}, \tau)}_{\text{locality-preserving}} + \lambda \left\| \boldsymbol{W} - \left( \sum_{j=1}^p \boldsymbol{W}_{jj}/p \right) I \right\|_{\text{F}}^2, \qquad (6)$$

where $\alpha$ is a hyperparameter that balances the trade-off between human alignment and preserving the local structure of a neural network's original representation space. We select values of $\alpha$ and $\lambda$ that give the lowest alignment loss via grid search; see details in Appx. A.2.

## 3.1 Downstream tasks

**Few-shot learning.** In general, few-shot learning (FS) is used to measure the transferability of neural network representations to different downstream tasks [86]. Here, we use FS to investigate whether the gLocal transform, as defined in §3, can improve downstream task performance and, hence, a network's transferability, compared to the original representation spaces. Specifically, we perform FS with and without applying the gLocal transforms. For all few-shot experiments, we use multinomial logistic regression, which has previously been shown to achieve near-optimal performance when paired with a good representation [88]. The regularization parameter is selected by $n_s$-fold cross-validation, with $n_s$ being the number of shots per class (more details in Appx. A.3).

**Anomaly detection.** Anomaly detection (AD) is a task where one has a collection of data considered "nominal" and would like to detect if a test sample is different from nominal data. For semantic AD tasks, e.g. nominal images contain a cat, it has been observed that simple AD methods using a pretrained neural network perform best [53, 4, 16]. In this work, we apply $k$-nearest neighbor AD to representations from a neural network, a method which has been found to work well [4, 53, 71]. We use the standard one-vs.-rest AD benchmark on classification datasets, where a model is trained using "one" training class as nominal data and performs inference with the full test set with the "rest" classes being anomalous [75].

## 4 Experimental results

In this section, we report experimental results for different FS and AD tasks. In addition, we analyze the effect of the gLocal transforms on both local and global similarity structures and report changes in representational alignment of image/text models for four human similarity judgment datasets. We start this section by introducing the different tasks and datasets and continue with the analyses.

### 4.1 Experimental setup

**CIFAR-100 coarse** The 100 classes in CIFAR-100 can be grouped into 20 semantically meaningful superclasses. Here, we use these superclasses as targets for which there exist 100 test images each.

**CIFAR-100 shift** simulates a distribution shift between the normal distribution of the training and testing images in CIFAR-100. For each of the 20 superclasses, there exist five subclasses. We use the first three subclasses for training and the last two subclasses for testing.

**Entity-13 and Entity-30** are datasets derived from ImageNet [17, 76]. They have been defined as part of the BREEDS dataset for subpopulation shift [77]. In BREEDS, ImageNet classes are grouped into superclasses based on a modified version of the WordNet hierarchy. Specifically, one starts at the *Entity* node of that hierarchy and traverses the tree in a breadth-first manner until the desired level of granularity (three steps from the root for Entity-13 and four for Entity 30). The classes residing at that level are considered the new coarse class labels and a fixed number of subclasses are sampled from the trees rooted at these superclass nodes — twenty for Entity-13 and eight for Entity-30. Through this procedure, Entity-13 results in more coarse-grained labels than Entity-30. The subclasses of each superclass are partitioned into source (training) and target (test) classes, introducing a subpopulation shift. For testing, we use all 50 validation images for each subclass.

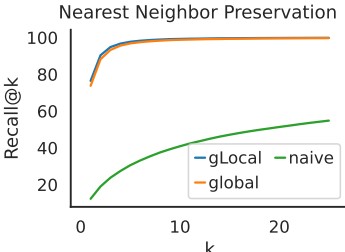

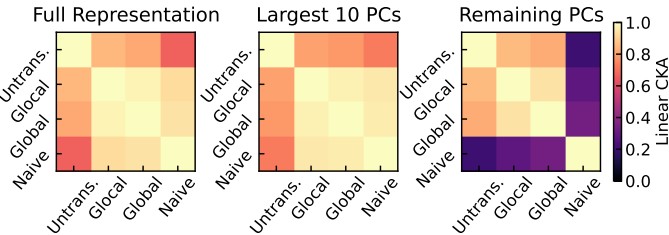

Figure 2: gLocal and global but not naive transforms preserve nearest neighbors in CLIP ViT-L representations. y-axis indicates the percentage of ImageNet validation images for which the closest image in the untransformed space is among the $k$ closest after transformation.

Figure 3: The top principal components (PCs) of gLocal and global representations of ViT-L on the ImageNet validation set resemble those of the naive transformed representation, indicating that they share similar global structure. The remaining PCs more closely match the untransformed representation. **Left:** CKA between full representations. **Middle:** CKA after setting singular values of each representation to zero for all but the largest 10 PCs. **Right:** CKA after setting singular values to zero for the largest 10 PCs, but retaining smaller PCs.

## 4.2 Impact of transforms on global and local structure

Our goal is to use human similarity judgments to correct networks' global representational structure without affecting local representational structure. Here, we study the extent to which our method succeeds at this goal. To quantify distortion of local structure, we first find the nearest neighbor of each ImageNet validation image among the remaining validation 49,999 images in the original representation space. We then measure the proportion of images for which the nearest neighbor in the untransformed representation space is among the closest $k$ images in the transformed representation space. As shown in Fig. 2, both the gLocal and global transforms generally preserve nearest neighbors, although the gLocal transform is slightly more effective, preserving the closest neighbor of 76.3% of images vs. 73.7% for the global transform. By contrast, the naive, unregularized transform preserves the closest neighbor in only 12.2% of images. For further intuition, we show the neighbors of anchor images in the untransformed, gLocal, and naively transformed representation spaces in Appx. C.

Whereas the local structure of gLocal and global representations closely resembles that of the original representation, the global structure instead more closely resembles the naive transformed representation. We quantify global representational similarity using linear centered kernel alignment (LCKA) [43, 14]. LCKA can be thought of as measuring the similarity between principal components (PCs) of two representations weighted by the amount of variance they explain; see further discussion in Appx. G. It thus primarily reflects similarity between the large PCs that define global representational structure. As shown in Fig. 3 (left), LCKA indicates that the gLocal/global representations are more similar to the naive transformed representation than to the untransformed representation, suggesting that the gLocal, global, and naive transforms induce similar changes in global structure. We further measure LCKA between representations obtained by setting all singular values to zero except for those corresponding to either largest 10 PCs or the remaining 758 PCs. LCKA between representations retaining the largest 10 PCs resembles LCKA between the full representations (Fig. 3 middle). However, when retaining only the remaining PCs, the gLocal/global representations are more similar to the untransformed representation than to the naive-transformed representation (Fig. 3 right).

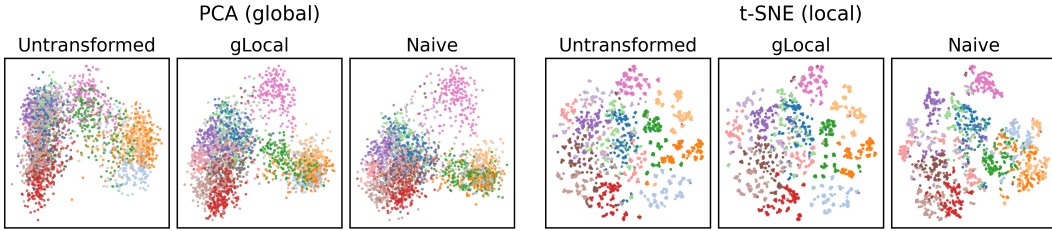

Figure 4: The gLocal transformed representation captures global structure of the naive transformed representation, as shown by PCA, but local structure of the untransformed representation, as shown by t-SNE with perplexity 10. Visualizations reflect embeddings of 10 images from each of the 260 Entity-13 ImageNet subclasses obtained from CLIP ViT-L and are colored by the superclasses. 2D embeddings are rotated to align with the gLocal representation using orthogonal Procrustes. The t-SNE fitting process is initialized using PCA; we pick the embedding with the lowest loss from 10 runs.

In Fig. 4, we visualize the effects of different transformations using PCA, which preserves global structure, and t-SNE, which preserves local structure. As described by Van der Maaten & Hinton [89], PCA "focus[es] on keeping the low-dimensional representations of dissimilar datapoints far apart," whereas t-SNE tries to "keep the low-dimensional representations of very similar datapoints close together." In line with the results above, the global structure of the gLocal representation revealed by PCA closely resembles that of the naive transformed representation, whereas the local structure of the gLocal representation revealed by t-SNE closely resembles that of the untransformed representation.

To complement these analyses, in Appx. B we explore what the alignment transforms alter about the global category structure of the model representations. Generally, categories within a single superordinate category move more closely together, while different superordinate categories move apart, but with sensible exceptions — e.g. food and drink move closer together.

## 4.3 Few-shot learning

Table 1: 5-shot FS results using the original or transformed representations. † indicates the highest accuracy for each dataset. Results are averaged over 5 runs.

| Model \ Transform | Entity-13 | | Entity-30 | | CIFAR100-Coarse | | CIFAR100 | | SUN397 | | DTD | |
|---|---|---|---|---|---|---|---|---|---|---|---|---|
| | original | gLocal | original | gLocal | original | gLocal | original | gLocal | original | gLocal | original | gLocal |
| CLIP-RN50 (WIT) | 63.99 | **67.96** | 57.86 | **59.80** | 44.27 | **47.43** | 38.77 | **39.59** | 57.21 | **58.79** | 54.32 | **54.95** |
| CLIP-ViT-L/14 (WIT) | 65.34 | **71.94**† | 66.92 | **69.97**† | 66.97 | **68.48** | 72.22 | **73.03** | 69.87 | **71.13** | 62.84 | **63.27** |
| CLIP-ViT-L/14 (LAION-400M) | 65.33 | **69.02** | 62.93 | **65.99** | 68.88 | **69.58** | **73.57** | 72.98 | 70.25 | **71.08** | **67.81** | 66.71 |
| CLIP-ViT-L/14 (LAION-2B) | 65.98 | **71.24** | 65.64 | **67.93** | 72.43 | **73.48**† | **79.01**† | 78.48 | 71.62 | **72.62**† | **69.49**† | 68.44 |

In this section, we examine the impact of the gLocal transform on few-shot classification accuracy on downstream datasets. We consider a standard *fine-grained* few-shot learning setup on CIFAR-100 [48], SUN397 [90], and the Describable Textures Dataset [DTD, 12], as well as a *coarse-grained* setup on Entity-{13,30} of BREEDS [77].

In coarse-grained FS, classes are grouped into semantically meaningful superclasses. This is a more challenging setting than the standard fine-grained scenario, for which there does not exist a superordinate grouping. In fine-grained FS, training examples are uniformly drawn from all (sub-)classes. For coarse-grained FS, we classify according to superclasses rather than fine-grained classes, and choose $k$ training examples uniformly at random for each superclass. This implies that not every subclass is contained in the train set if the number of training samples is smaller than the number of subclasses. On Entity-{13,30}, superclasses between train and test sets are guaranteed to be disjoint due to a subpopulation shift. To achieve high accuracy, models must consider examples from unseen members of a superclass similar to the few examples it has seen. Task performance is dependent on how well the partial information contained in the few examples of a superclass can be exploited by a model to characterize the entire superclass with its subclasses. Hence, global similarity structure is more crucial than local similarity structure to perform well on this task. We calculate classification accuracy across all available test images of all subclasses, using coarse labels.

We find that transforming the representations via gLocal transforms substantially improves performance over the untransformed representations across both coarse-grained and fine-grained FS tasks for all image/text models considered (Tab. 1). For CLIP models trained on LAION, however, we do not observe improvements for CIFAR-100 and DTD, which are the most fine-grained datasets. For ImageNet models, the gLocal transform improves performance on Entity-{13,30}, but has almost no impact on the performance for the remaining datasets (see Appx. D.1).

## 4.4 Anomaly detection

Here, we evaluate the performance of representations on $k$-nearest neighbor AD with and without using the gLocal transform. AD methods typically return an anomaly score for which a detection threshold has to be chosen. Our method is evaluated using each training class as the nominal class and we report the average AUROC. Following §3.1, we compute the representations for each normal sample of the training set and then evaluate the model with representations from the test set. We set $k = 5$ for our experiments but found that performance is fairly insensitive to the choice of $k$ (see Appx. D.2). For measuring the distance between representations, we use cosine similarity.

In Tab. 2, we show that the gLocal transform substantially improves AD performance across all datasets considered in our analyses, for almost every image/text model. However, as in the few-shot setting, we observe no improvements over the untransformed representation space for ImageNet models (see Appx. D.2). In Tab. 3, we further investigate performance on distribution shift datasets,

where improvements are particularly striking. Here, global similarity structure appears to be crucial for generalizing between the superclasses. Tab. 2 additionally reports current state-of-the-art (SOTA) results on the standard benchmarks; SOTA results are not available for the distribution shift datasets. SOTA approaches generally use additional data relevant to the AD task, such as outlier exposure data or textual supervision for the normal class [53, 13, 36], whereas we use only human similarity judgments. Our transformation also works well in non-standard AD settings (see Appx. D.2).

Table 2: One-vs-rest nearest neighbor based AD results; with and without transformation. † indicates the highest accuracy for each dataset.

| Model \ Transform | CIFAR10 | | CIFAR100 | | CIFAR100-Coarse | | ImageNet30 | | DTD | |
|---|---|---|---|---|---|---|---|---|---|---|
| | original | gLocal | original | gLocal | original | gLocal | original | gLocal | original | gLocal |
| CLIP-RN50 (WIT) | 89.44 | **91.19** | 90.83 | **92.92** | 86.47 | **89.29** | 98.89 | **99.05** | 90.67 | **92.29** |
| CLIP-ViT-L/14 (WIT) | 95.14 | **98.16** | 91.41 | **97.19** | 88.5 | **95.83** | 98.91 | **99.75**† | 92.02 | **94.9** |
| CLIP-ViT-L/14 (LAION-400M) | **98.8** | 98.39 | **98.66** | 98.53 | **97.86** | 97.77 | 99.51 | **99.69** | 95.54 | **96.44** |
| CLIP-ViT-L/14 (LAION-2B) | 98.97 | **99.11**† | 98.76 | **98.97**† | 98.05 | **98.5**† | 99.29 | **99.74** | 94.87 | **96.78**† |
| SOTA | 99.6 [53] | | - | | 97.34 [13] | | 99.9 [53] | | 94.6 [53] | |

Table 3: One-vs-rest AD with a class distribution shift between train and test sets; with and without transformation. † indicates the highest accuracy for each dataset.

| Model \ Transform | Entity-13 | | Entity-30 | | Living-17 | | Nonliving-26 | | Cifar100-shift | |
|---|---|---|---|---|---|---|---|---|---|---|
| | original | gLocal | original | gLocal | original | gLocal | original | gLocal | original | gLocal |
| CLIP-RN50 (WIT) | 90.22 | **92.03** | 91.64 | **93.71** | **94.62** | 93.28 | 87.49 | **91.09** | 76.27 | **80.33** |
| CLIP-ViT-L/14 (WIT) | 88.54 | **93.23**† | 91.31 | **95.53** | 97.31 | **97.7**† | 84.73 | **92.53** | 73.69 | **87.11** |
| CLIP-ViT-L/14 (LAION-400M) | 90.79 | **92.71** | 92.49 | **95.04** | **96.56** | 96.32 | 90.09 | **93.18** | 91.09 | **92.7** |
| CLIP-ViT-L/14 (LAION-2B) | 90.33 | **93.08** | 92.1 | **95.48**† | 96.96 | **97.37** | 88.82 | **93.54**† | 89.73 | **93.94**† |

## 4.5 Representational alignment

We have shown above that the gLocal transform provides performance gains on FS and AD tasks. Here, we examine whether these performance gains come at the cost of alignment with human similarity judgments as compared to the naive transform, which does not preserve local structure. Thus, we examine the impact of the gLocal transform on human alignment, using the same human similarity judgment datasets evaluated in Muttenthaler et al. [61] plus an additional dataset.[4] Specifically, we perform representational similarity analysis [RSA; 46] between representations of two image/text models — CLIP RN50 and CLIP ViT-L/14 — and human behavior for four different human similarity judgment datasets [31, 65, 66, 10, 41]. RSA is a method for comparing neural network representations to representations obtained from human behavior [46]. In RSA, one first obtains representational similarity matrices (RSMs) for the human behavioral judgments and for the neural network representations (more details in Appx. E). These RSMs measure the similarity between pairs of examples according to each source. As in previous work [10, 41, 61], we use the Spearman rank correlation coefficient to quantify the similarity of these RSMs. We find that there is almost no trade-off in representational alignment for the gLocal transform compared to the naively transformed representations (see Tab. 4). Hence, the gLocal transform can improve representational alignment while preserving local similarity structure.

In Fig. 5, we further demonstrate how the representational changes introduced by the linear and gLocal transforms lead to greater alignment with human similarity judgments by visualizing the RSMs on each dataset. The global similarity structure captured by the RSMs is qualitatively identical between naive and gLocal transforms, and both of these transforms lead to RSMs that closely resemble human RSMs (see Fig. 5). Human similarity judgments for data from Hebart et al. [31] were collected in the form of triplet odd-one-out choices. Therefore, we used VICE [60] — an approximate Bayesian method for inferring mental representations of object concepts from human behavior — to obtain an RSM for those judgments. Human RSMs are sorted into five different concept clusters using $k$-means for datasets from King et al. [41] and Cichy et al. [10] and using the THINGS concept hierarchy [30] for RSMs from Hebart et al. [31]. A visualization for RSMs obtained from CLIP RN50 representations and a more detailed discussion on RSA can be found in Appx. E.

---

[4]Human similarity judgments were collected by either asking participants to arrange natural object images from different categories on a computer screen [66, 10, 41] or in the form of triplet odd-one-out choices [31].

Table 4: The gLocal transform yields both a high degree of alignment with datasets of human similarity judgments and good performance on FS/AD tasks. Performance on human similarity datasets is measured as odd-one-out accuracy on a held-out test set for THINGS or Spearman's $\rho$ for the other three datasets, using either the original representations, the naively transformed representations, or representations transformed via the gLocal transform. For 5-shot FS and AD, we report the average performance across all tasks in Tab. {1, 2, 3}.

| Dataset \ Transform | CLIP-RN50 (WIT) | | | CLIP-ViT-L/14 (WIT) | | | CLIP-ViT-L/14 (LAION-2B) | | |
|---|---|---|---|---|---|---|---|---|---|
| | original | naive | gLocal | original | naive | gLocal | original | naive | gLocal |
| *Human similarity datasets:* | | | | | | | | | |
| Hebart et al. [31] | 52.78% | 59.92% | 59.89% | 46.71% | 60.13% | 60.05% | 47.50% | 60.47% | 60.38% |
| King et al. [41] | 0.386 | 0.650 | 0.645 | 0.355 | 0.638 | 0.637 | 0.292 | 0.620 | 0.613 |
| Cichy et al. [10] | 0.557 | 0.721 | 0.716 | 0.363 | 0.732 | 0.732 | 0.395 | 0.735 | 0.718 |
| Peterson et al. [65, 66] | 0.364 | 0.701 | 0.705 | 0.260 | 0.688 | 0.688 | 0.314 | 0.689 | 0.660 |
| *Downstream tasks:* | | | | | | | | | |
| 5-shot FS (avg.) | 53.05% | 41.20% | **55.01%** | 67.71% | 47.62% | **69.91%** | 71.10% | 50.71% | **72.27%** |
| Anomaly detection (avg.) | 89.65% | 90.75% | **91.52%** | 90.16% | 92.79% | **95.19%** | 94.79% | 93.09% | **96.65%** |

# 5 Discussion

Although neural networks achieve near-human-level performance on a variety of computer vision tasks, they may not optimally capture *global* object similarities. By contrast, humans represent concepts using rich semantic features — including superordinate categories and other global constraints — for performing object similarity judgments [66, 41, 10, 31, 60]. These representations may contribute to humans' strong generalization capabilities [21, 24]. Here, we investigated the impact of aligning neural network representations with human similarity judgments on different FS and AD tasks.

We find that naively aligning neural network representations, without regularizing the learned transformations to preserve structure in the original representation space, can impair downstream task performance. However, our *gLocal transform*, which combines an *alignment loss* that optimizes for representational alignment with a *local loss* that preserves the nearest neighbor structure from the original representation, can substantially improve downstream task performance while increasing representational alignment. The transformed representation space transfers surprisingly well across different human similarity judgment datasets and achieves almost equally strong alignment as the naive approach [61], indicating that it captures human notions of object similarity. In addition, it substantially improves downstream task performance compared to both original and naively aligned representations across a variety of FS and AD tasks. The gLocal transform yields state-of-the-art (SOTA) performance on the CIFAR-100 coarse AD task, and approaches SOTA on other AD benchmarks.

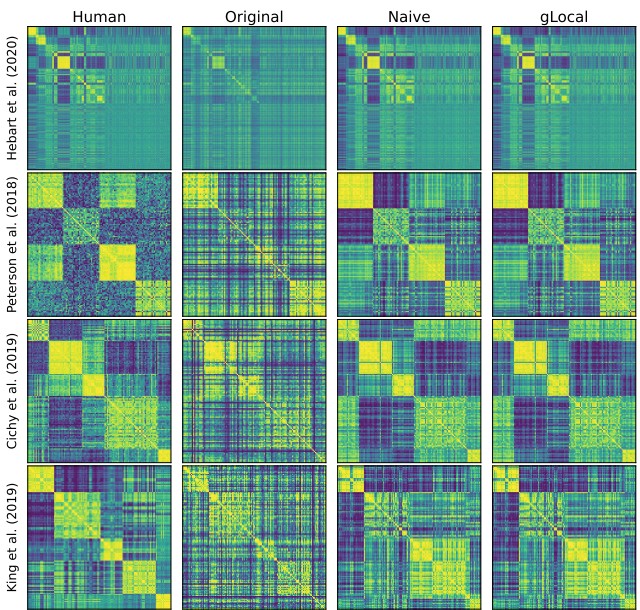

Figure 5: RSMs for human behavior and CLIP ViT-L/14 [WIT; 70] for four different human similarity judgment datasets [31, 66, 10, 41]. We contrast RSMs obtained from the network's original representations (second column), the naively aligned representations [61] (third column), and the representations + gLocal transform (rightmost column) against RDMs directly constructed from human similarity judgments (leftmost column). Yellower colors indicate greater similarity; bluer colors indicate greater dissimilarity.

**Ablations.** In addition, we show that the gLocal transform can readily be used in combination with other approaches that are specifically designed for few-shot learning with pretrained representations of image/text models — such as Tip-Adapter [93]. We observe considerable improvements compared to using Tip-Adapter without applying the gLocal transform (see Appx. D.1). This indicates that

methods designed to improve the transferability of pretrained representations without incorporating knowledge about human global similarity structure can additionally benefit from the gLocal transform. Using a human-adversarial triplet dataset, where each odd-one-out choice is determined to be an object that is different from the human choice, for optimizing the gLocal transform either yields no change in a model's representation space and thus no difference in downstream task performance or changes a model's representation space in a way that substantially deteriorates its downstream task performance (see Appx. D.4). These observations corroborate our findings and suggest that the benefits of the gLocal transform most likely stem from including an inductive bias about global object similarity.

**Limitations.** Our work has some limitations. First, as we show in Appx. D, the gLocal transform fails to consistently improve downstream task performance on ImageNet. We conjecture that the gLocal transform can succeed only if representations capture the concepts by which human representations are organized, and ImageNet representations may not. Second, human similarity judgments are more expensive to acquire than other forms of supervision, and there may be important human concepts that are captured neither by the 1854 images we use for alignment nor by the pretrained representations.

**Conclusion.** Our results imply that even with hundreds of millions of image/text pairs, image/text contrastive learning does not learn a representation space with human-like global organization. However, since the gLocal transform successfully aligns contrastive representations' global structure using only a small number of images, these representations do seem to have a pre-existing representation of the concepts by which human representations are globally organized. Why does this happen? One possibility is that image/text pairs do not provide adequate global supervision, and thus contrastive representation learning is (near-)optimal given the data. Alternatively, contrastive learning may not incorporate signals that exist in the data into the learned representation because it imposes only local constraints. Previous work has shown that t-SNE and UMAP visualizations reflect global structure only if they are carefully initialized [42]. Given the similarity between contrastive representation learning and t-SNE/UMAP [15] and the known sensitivity of contrastive representations to initialization [52], it is plausible contrastive representations also inherit their global structure from their initialization. Although our gLocal transform provides a way to perform post-hoc alignment of representations from image/text models using human similarity judgments, there may be alternative initialization strategies or objectives that can provide the same benefits during training, using only image/text pairs.

## Acknowledgements

LM, LL, JD, and RV acknowledge funding from the German Federal Ministry of Education and Research (BMBF) for the grants BIFOLD22B and BIFOLD23B. LM acknowledges support through the Google Research Collabs Programme. We thank Pieter-Jan Kindermans for helpful comments on an earlier version of the manuscript.

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

# A   Experimental details

## A.1   Model features

We extract penultimate layer features of four different ImageNet-models — AlexNet [47], VGG-16 [85], ResNet-18, and ResNet-50 [28] — and image encoder features of four different image/text models — CLIP RN50 and CLIP ViT-L/14 trained on WIT [70]; CLIP ViT-L/14 trained on Laion-400M [81] and Laion-2B [82] respectively. For extracting the model features, we use the Python library `thingsvision` [59].

## A.2   gLocal probing

To optimize the gLocal transforms, we use standard SGD with momentum and perform cross-validation according to the procedure proposed in Muttenthaler et al. [61]. For finding the optimal gLocal transform, we perform an extensive grid search over four different hyperparameter values — the learning rate, $\eta$, the strength of the regularization term $\lambda$, the global-local trade-off parameter $\alpha$, and the temperature parameter, $\tau$, used in the softmax expression for the local contrastive loss term (see Eq. 5). Specifically, we perform an extensive grid search over the Cartesian product of the following sets of hyperparameters:

- $\eta \in \{0.0001, 0.001, 0.01, 0.1\}$,
- $\lambda \in \{0.01, 0.1, 1.0, 10.0\}$,
- $\alpha \in \{0.05, 0.1, 0.25, 0.5, 1.0\}$,
- $\tau \in \{0.1, 0.25, 0.5, 1.0\}$.

We use the same $\eta$ and $\lambda$ grids for global probing. We use `PyTorch` [63] for implementing the probes and PyTorch lightning to accelerate training. We choose the gLocal transform that achieves the lowest alignment loss (see alignment term in Eq. 6). Among the values in the above grid, we find that a combination of $(\alpha = 0.1, \lambda = 0.1, \eta = 0.001)$ yields the lowest alignment loss/highest probing odd-one-out accuracy for both CLIP RN50 (WIT) and CLIP ViT-L/14 (WIT) (see Fig. 6). A combination of $(\alpha = 0.25, \lambda = 0.1, \eta = 0.001)$ gives the second lowest alignment loss/highest probing odd-one-out accuracy for CLIP RN50 (WIT) and CLIP ViT-L/14 (WIT).

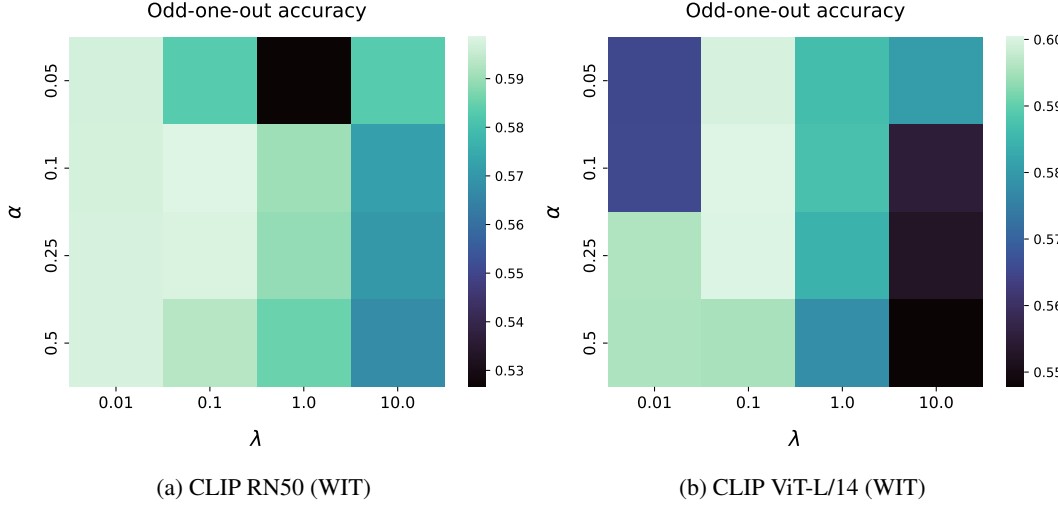

(a) CLIP RN50 (WIT)          (b) CLIP ViT-L/14 (WIT)

Figure 6: Among all hyperparameter combinations considered in our grid search, a combination of $(\alpha = 0.1, \lambda = 0.1, \eta = 0.001)$ for Eq. 6 in §3 yields the best odd-one-out accuracy on a held-out test set for both CLIP RN50 and CLIP ViT-L/14 (WIT).

For each $(\alpha, \lambda)$ combination we select that combination with the best probing odd-one-out accuracy on a held-out test among the set of possible learning rate, $\eta$, and temperature value, $\tau$, combinations determined by the above grid. We observe that $\eta = 0.001$ generally gives the best results across the different $(\alpha, \lambda)$ combinations, whereas performance is fairly insensitive to the value of $\tau$. Since neither $(\alpha = 0.1, \lambda = 0.1)$ nor $(\alpha = 0.25, \lambda = 0.1)$ are values at the edges of the hyperparameter

grid, it is plausible to assume that both the contrastive local loss and the regularization term in Eq. 6 in §3 are necessary to obtain a transformation that leads to a *best-of-both-worlds* representation.

Although our goal has been to find a transform that induces both increased representational alignment and improved downstream task performance, we considered $\alpha = 1.0$ to examine whether downstream task performance can potentially be improved by excluding the alignment loss. Note that $\alpha = 1.0$ causes the optimization process to ignore the alignment loss. Unsurprisingly we did not find that to be the case. We remark that minimizing both the local contrastive loss and the regularization preserves the local similarity structure of the original representation space but does not inject any additional information into the representations. Moreover, it is non-trivial to choose a transform that works well across all downstream tasks without including the alignment loss. Therefore, we exclude $\alpha = 1.0$ in Fig. 6.

**Compute**. We used a compute time of approximately 400 hours on a single Nvidia A100 GPU with 40GB VRAM for all linear probing experiments — including the hyperparameter sweep. The computations were performed on a standard, large-scale academic SLURM cluster.

### A.3   Few-shot learning

Here, we use $n_s$-fold cross-validation for finding the optimal $\ell_2$-regularization parameter, where $n_s$ refers to the number of shots per class. We select the parameter from the following set of values, $\{1e{+}6, 1, 1e{+}5, 1e{+}4, 1e{+}3, 1, 1e{+}2, 1e{+}1, 1, 1e{-}1, 1e{-}2, 1e{-}3, 1e{-}4\}$. We use the `scikit-learn` [64] implementation of (multinomial) logistic regression and refit the regression after selecting the optimal regularization parameter.

**Compute**. We used a compute time of approximately 5600 CPU-hours of 2.90GHz Intel Xeon Gold 6326 CPUs for all few-shot experiments. Computations were performed on a standard, large-scale academic SLURM cluster.

### A.4   Anomaly Detection

In this section, we outline our anomaly detection experimental setting in more detail. In the anomaly detection settings that we consider in our analyses *normal/anomaly* classes are determined via the original classes in the data. Here, each of the original classes is once selected as a normal class with the remaining classes being anomalous and, vice versa, each class in the data is once selected as an anomalous class with the other classes being normal. After embedding the training images from either the normal or the anomalous class in a model's representation space, at inference time a model must classify whether a new image belongs to the normal data or whether it deviates from it and is thus considered an anomaly. For each example in the test set, a model yields an anomaly score where higher scores indicate more probability of an example being anomalous. Using the binary anomaly labels and the anomaly scores for each of the examples, we can then compute the *area-under-the-receiver-operating-characteristic-curve* (AUROC) to quantify the performance of the model.

**One-vs-rest**. Given a dataset (e.g., CIFAR-10) with $C$ classes, one class (e.g., "airplane") is chosen to be the normal class and the remaining $C - 1$ classes of the dataset are considered anomalies. Each of the $C$ classes is once selected as a normal class and the AUROC is averaged across the classes.

**Leave-one-out (LOO)**. In contrast to the "one-vs-rest" setting, in LOO we define one class of the dataset as an anomaly and the remaining classes as normal. Similarly to the "one-vs-rest" setting, this results in $C$ evaluations for a dataset with $C$ classes.

In both "one-vs-rest" and LOO AD settings, we evaluate model representations in the following way: First, we compute the representations $X_{\text{train}}$ of the normal samples in the train set. Then, we compute the representations of all test set examples $X_{\text{test}}$. For each test set representation, we compute the cosine similarity to all normal train set representations, $X_{\text{train}}$, and select the $k$ nearest neighbor samples that have the highest cosine similarity.

The anomaly score of a test set representation is then defined as the average cosine distance to the $k$ nearest train representations. $k$ is a hyperparameter that determines the number of nearest neighbors over which the anomaly score is computed. We choose $k = 5$ for our experiments but show that performance is fairly insensitive to the value of $k$ (see Tab. 12).

**Compute**. For all AD experiments, we used a compute time of approximately 20 hours on a single Nvidia A100 GPU with 40GB VRAM. Computations were performed on a standard, large-scale academic SLURM cluster.

# B  What changes in the global structure of the representations after alignment?

In this section, we attempt to build some intuitions for how the global structure of the representations changes after alignment. To do so, we analyze the movements of the representations of items and superordinate categories in the THINGS dataset. Specifically, we compute cosine differences between the CLIP-ViT-L/14 representations of each pair of items in THINGS and then compute how these distances change under the transforms.

We show the pairs of items that change the most in distance in Table 5. Items that are semantically related, like "curry" and "scrambled egg", tend to move closer together, and therefore have transformed distances that are smaller than their original distance. By contrast, items like "handcuff" and "stethoscope", which are semantically unrelated but perhaps have some slight visual similarity, tend to move farther apart. The distance changes under the gLocal transforms are correlated with, though generally less varied than, those under the naively-aligned transform.

To more broadly analyze the change in global structure, we then look at how distances between pairs of items change within and across superordinate categories (the top-down categories from THINGS). We show the results in Fig. 7. Under the naively aligned transform, the items within each superordinate category tend to move slightly closer together — the diagonal is slightly blue — while the items from different categories tend to move substantially farther apart — the off-diagonal is mostly red. That is, the representations are broadly moving in a way that reflects the overall human semantic organization of the categories.

There are a few notable standouts: the categories of drink, food, plant, and animal change particularly much, and in particularly interesting ways. These categories each move much farther relative to all other categories (such as tool or musical instrument) than those other categories move relative to each other. This perhaps reflects the particular semantic salience of food, drink, plants, and animals from a human perspective. Furthermore, food and drink are one of the few pairs of superordinate categories between which distances actually decrease after the transform, presumably reflecting the strong semantic ties between these categories. Similarly, animals move less far from plants than from any other category, perhaps reflecting the fact that the animate/inanimate distinction is one of the strongest features in human semantic representations [73].

Under the gLocal transform, the pattern of changes is strongly correlated with the naively aligned transform ($r = 0.96$, $p \leq 10^{-16}$). However, in keeping with the regularization, the magnitude of the changes varies less.

Table 5: Distances between pairs of individual items from THINGS, ranked by the relative change in cosine distance from before to after naive alignment (normalized by original distance). The top items move much closer together under naive alignment, while the bottom ones move much farther apart. (All results are from CLIP-ViT-L/14.)

| Item 1 | Item 2 | original dist. | naively aligned dist. | gLocal dist. |
|--------|--------|---------------|----------------------|--------------|
| curry | scrambled egg | 0.303120 | 0.005276 | 0.401019 |
| otter | warthog | 0.305242 | 0.005530 | 0.382150 |
| parfait | spaghetti | 0.457553 | 0.009115 | 0.540346 |
| otter | rhinoceros | 0.327497 | 0.006530 | 0.456641 |
| | | $\vdots$ | | |
| stethoscope | wheat | 0.263908 | 1.284535 | 0.935891 |
| grass | wallet | 0.277866 | 1.347424 | 1.056572 |
| cat | traffic light | 0.285151 | 1.378671 | 0.981944 |
| handcuff | sugar cube | 0.272936 | 1.308337 | 0.904380 |

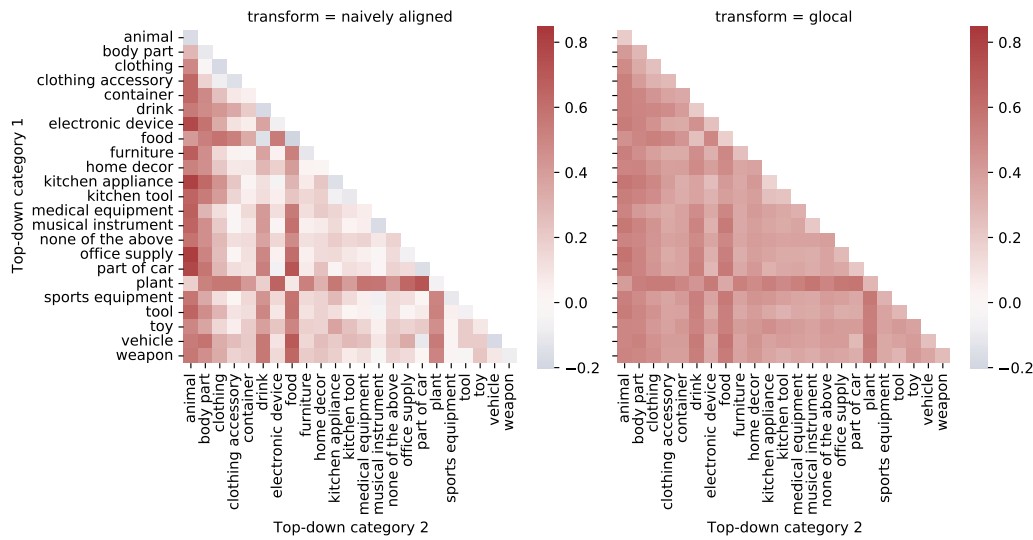

Figure 7: How does the global structure of the representations change after alignment? Here, we analyze the movements of the representations of pairs of items from different superordinate categories from the THINGS dataset. The squares on the diagonal indicate the change in distance between items within a superordinate category, while the squares off the diagonal indicate changes between pairs of items from the corresponding pair of superordinate categories. A red color indicates the items from the categories move farther apart from each other after alignment, blue indicates moving closer together. Generally, items within a superordinate category move slightly closer together under naive alignment, while those in different categories move farther apart. A similar overall pattern is reflected in both the naively-aligned transform (left) and gLocal (right) ones, though under gLocal alignment there is a greater overall spreading of the representations. (All results are from CLIP-ViT-L/14.)

## C   Visualization of neighboring images

To provide further insight into the difference between the effects of the naive and gLocal transforms, in Fig. 8 we visualize the neighbors of nine anchor images. In order to show a diverse set of images, we pick the nearest neighbors in the CLIP ViT-L/14 (WIT) embedding space subject to the constraint that each neighbor comes from a different class from the original images and the nearer neighbors. In accordance with the results in §4.2, we find that the neighbors in the untransformed and gLocal spaces are generally similar, whereas neighbors in the naive representation space are frequently different. The naive transform appears to discard all non-semantic properties of images, whereas the untransformed and gLocal representation spaces are sensitive to pose, color, and numerosity. In cases where the closest neighbor differs between the naive and gLocal representations (third and fourth row), the neighbors in the gLocal representation are arguably better matches to the anchor.

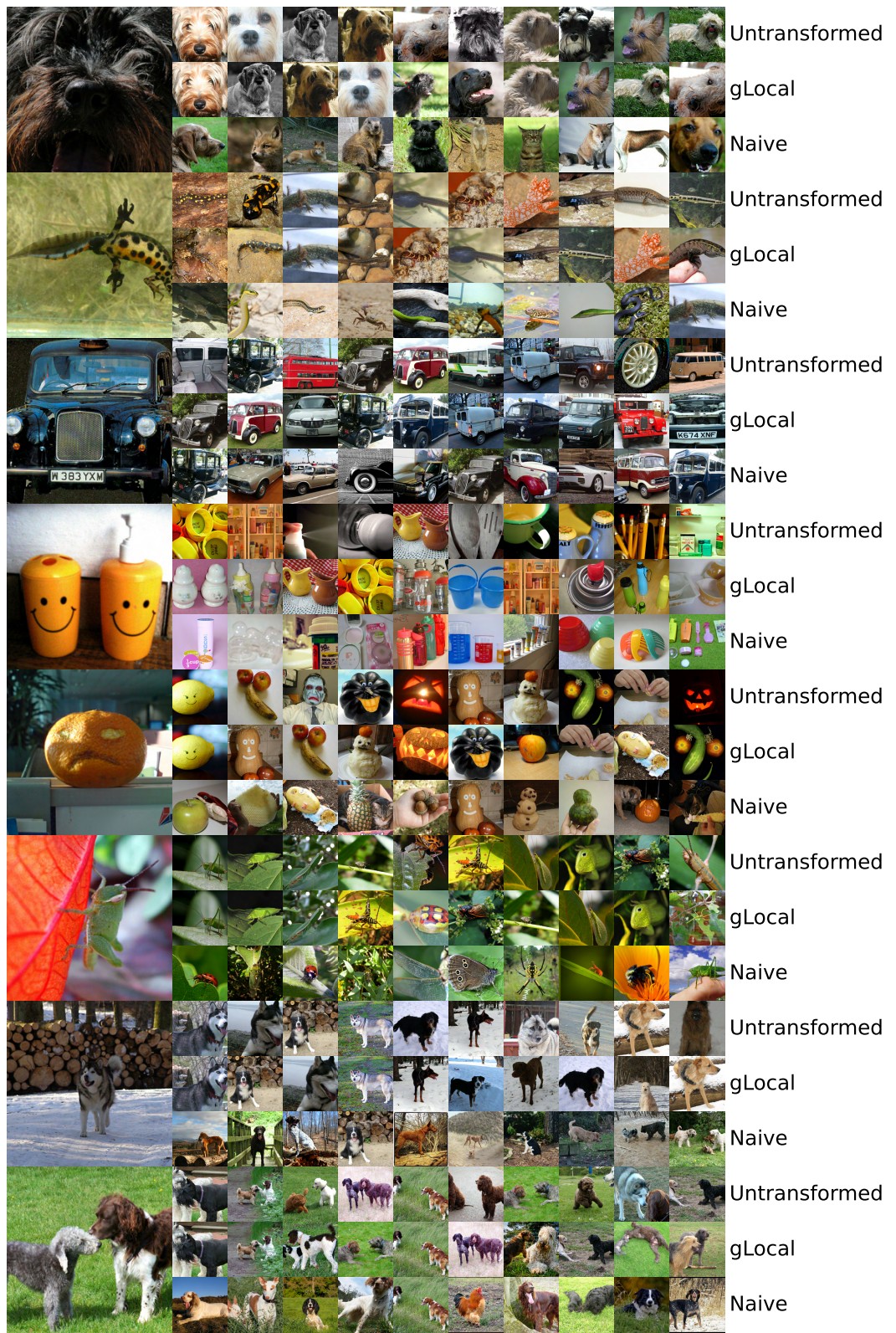

Figure 8: Comparison of neighbors in the ImageNet validation set for representations with different transforms. We visualize the 10 closest images subject to the constraint that each comes from a unique class. The anchor images are shown in the leftmost column. The three rows corresponding to each anchor image show their nearest neighbors in the untransformed, gLocal transformed, and naively transformed representations.

# D Additional results on downstream tasks

In this section, we provide additional few-shot learning and anomaly detection results for all ImageNet and image/text models that we considered in our analyses (see §A.1). We start this section by demonstrating a strong relationship between the performances of the different downstream tasks.

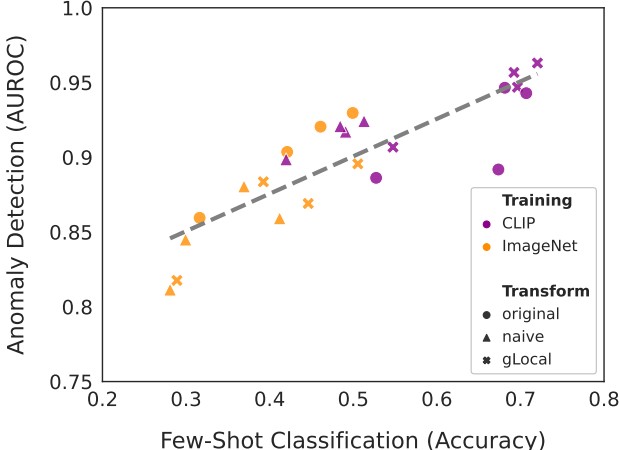

Figure 9: Here, we show anomaly detection AUROC averaged across all tasks reported in Tab. {2, 3} as a function of the average 5-shot classification performance for all ImageNet and CLIP models (see §A.1), using either the original representations or the representations transformed via the naive or gLocal transformations.

**Downstream task relationship**. We observe a strong positive relationship ($r = 0.8524, p \leq 10^{-6}$) between the average few-shot learning and the average anomaly detection performance for all ImageNet and image/text models that we considered in our analyses (see Fig. 9). This observation holds for both the original representation space and the representations transformed via the naive or gLocal transformations. This indicates that both downstream tasks require similar representations for similarly strong performance.

## D.1 Few-shot learning

In the following section, we show additional few-shot learning results. Specifically, we report 5-shot performance of ImageNet models and show few-shot results as a function of the number of samples used during fitting.

**Results for ImageNet models**. In Tab. 6 we report additional 5-shot results for ImageNet models. The gLocal transforms improve few-shot accuracy on Entity-{13,30} from BREEDS but the impact on few-shot performance is either inconsistent or negative for CIFAR-100 coarse, CIFAR-100, SUN397, and DTD of which the latter three are more fine-grained datasets than the other three.

Table 6: 5-shot FS results using the original or transformed representations.

| Model \ Transform | Entity-13 original | Entity-13 gLocal | Entity-30 original | Entity-30 gLocal | CIFAR100-Coarse original | CIFAR100-Coarse gLocal | CIFAR100 original | CIFAR100 gLocal | SUN397 original | SUN397 gLocal | DTD original | DTD gLocal |
|---|---|---|---|---|---|---|---|---|---|---|---|---|
| AlexNet | 37.49 | **42.13** | **27.73** | 26.93 | **32.33** | 31.44 | **28.70** | 23.29 | **26.43** | 19.12 | **37.05** | 30.54 |
| ResNet-18 | 56.71 | **58.19** | **52.94** | 51.27 | **42.97** | 41.84 | **38.12** | 35.92 | **37.26** | 34.52 | **48.53** | 45.90 |
| ResNet-50 | 49.27 | **59.18**† | 50.16 | **54.11**† | **50.99**† | 49.48 | **47.99**† | 44.75 | **47.17**† | 44.26 | **54.02**† | 51.49 |
| VGG-16 | 51.78 | **56.49** | 44.69 | **45.97** | **39.08** | 37.03 | **34.54** | 28.08 | **37.12** | 29.89 | **45.39** | 38.01 |

**Results for Tip-Adapter [93]**. To investigate whether the gLocal transforms improve downstream task performance of image/text models in few-shot learning settings in combination with techniques that were specifically designed for this task, we evaluate the CLIP models combined with Tip-Adapter [93]. Tip-Adapter is designed to produce effective few-show classifiers from pretrained image/text models by linearly combining the outputs of two modules:

(i) a *zero-shot classifier*, where the similarity of the input in embedding space with the embedded textual description of each class determines the output

(ii) a *key-value cache model* that computes its output by summing the one-hot labels of all stored few-shot data, weighted by the similarities of the associated image embeddings to the embedded imput sample

For the evaluation of gLocal transforms, we transform both the text embedding and any image embedding involved in the inference part. The hyper-parameters that we use are $\alpha = 1$ and $\beta = 5.5$ which have been identified as optimal for ImageNet classification by Zhang et al. [93].

We observe that even in conjunction with a method specifically designed for few-shot learning, gLocal transforms improve few-shot accuracy in most scenarios (see Tab. 7), in particular on Entity-{13,30} from BREEDS. Furthermore, the performance increase appears to be stronger for the CLIP models trained on the OpenAI WIT datasets compared to LAION (see Fig. 10). For the smaller of the two LAION datasets [LAION-400M; 81], the effect of the gLocal transform on CLIP with Tip-Adapter is negligible compared to its effect on models trained on the larger LAION-2B [82] or OpenAI's WIT dataset [70].

Table 7: 5-shot FS results using the original or transformed representations in combination with Tip-Adapter.

| Model \ Transform | Entity-13 original | Entity-13 gLocal | Entity-30 original | Entity-30 gLocal | CIFAR100-Coarse original | CIFAR100-Coarse gLocal | CIFAR100 original | CIFAR100 gLocal | SUN397 original | SUN397 gLocal | DTD original | DTD gLocal |
|---|---|---|---|---|---|---|---|---|---|---|---|---|
| CLIP-RN50 (WIT) | 66.29 | **69.95** | 59.24 | **61.76** | 31.41 | **33.69** | 38.53 | **44.82** | 53.42 | **55.75** | **47.98** | 47.91 |
| CLIP-ViT-L/14 (WIT) | 68.17 | **77.15**$^\dagger$ | 67.50 | **73.52**$^\dagger$ | 55.59 | **66.73** | 45.79 | **64.51** | 64.77 | **69.80** | 57.96 | **59.72** |
| CLIP-ViT-L/14 (LAION-400M) | **72.87** | 72.04 | **69.60** | 67.21 | **75.07** | 71.38 | **69.65** | 68.27 | **71.24**$^\dagger$ | 70.30 | **68.96**$^\dagger$ | 65.69 |
| CLIP-ViT-L/14 (LAION-2B) | 73.19 | **75.54** | 70.50 | **71.79** | **77.45**$^\dagger$ | 76.81 | 71.87 | **73.33**$^\dagger$ | 70.24 | **71.17** | **66.10** | 64.43 |

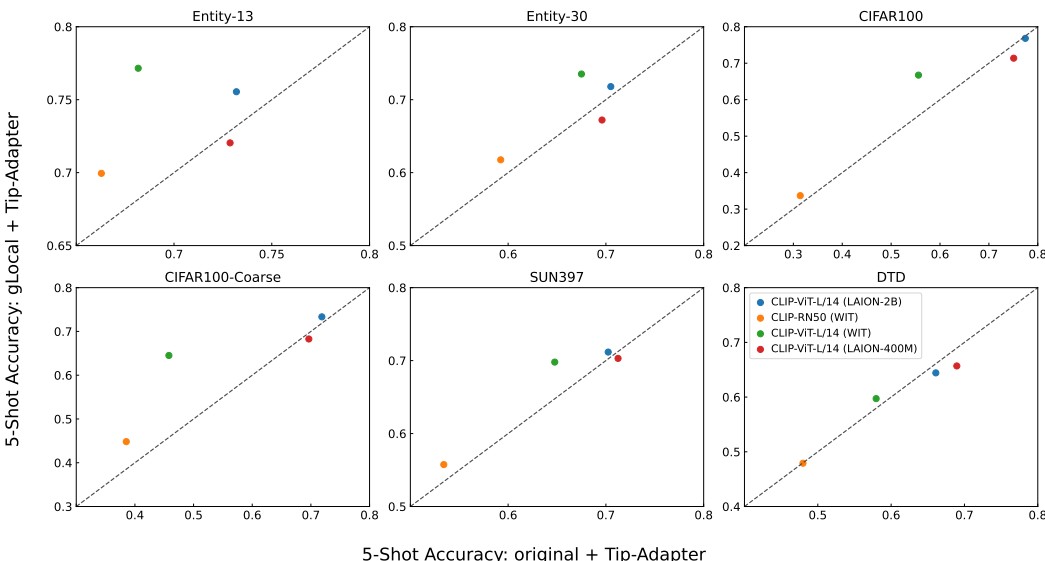

Figure 10: 5-shot FS results using the original or transformed representations in combination with Tip-Adapter.

**Effect of transforms for different numbers of training samples**. When varying the number of training samples for the few-shot experiments described in §4.3 we observe consistent improvements of the gLocal transforms across shots. Excluding the high-variance setting of 2-shot learning, we either find stable improvements in accuracy for image/text models, or a downward trend for ImageNet models on some tasks. This corroborates our findings from §4.3. Results appear to be robust to changes in the training set size, in particular for the CLIP models. Yet, we observe the most substantial benefits in low data regimes. See Fig. 11 for more details.

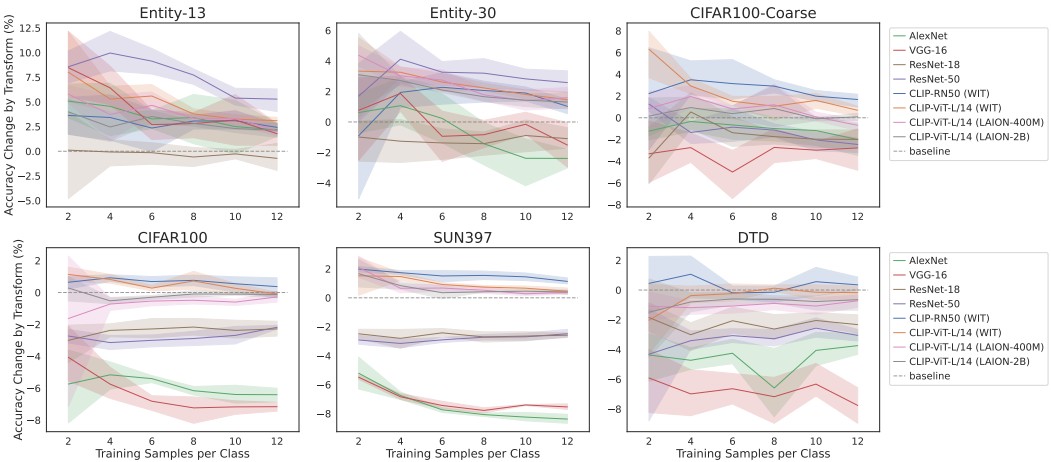

Figure 11: Change in average accuracy for different numbers of training samples per super-class (top row) or (sub-)class (bottom row) used for few-shot learning. Error bands depict 95% Confidence Intervals (CIs), computed over 5 different runs.

**Few-shot learning on ImageNet-1K**. In addition to the few-shot learning results that we presented in §4.3, here we report few-shot classification results on ImageNet-1K [17]. For both the original and gLocal-transformed CLIP representations, we trained a linear classifier with different numbers of shots, $k$, i.e., the number of training examples per class. We find that the gLocal-transformed representation achieves better ImageNet validation set accuracy compared to the original representation for both CLIP ViT-L/14 (WIT) and CLIP ViT-L/14 (LAION-2B) in small training data settings where $k > 5$. We don't observe a significant difference between the two representation spaces for CLIP RN50 (WIT) and CLIP ViT-L/14 (LAION-400 M) which generally perform worse than the other two models across all training settings (see Fig. 12).

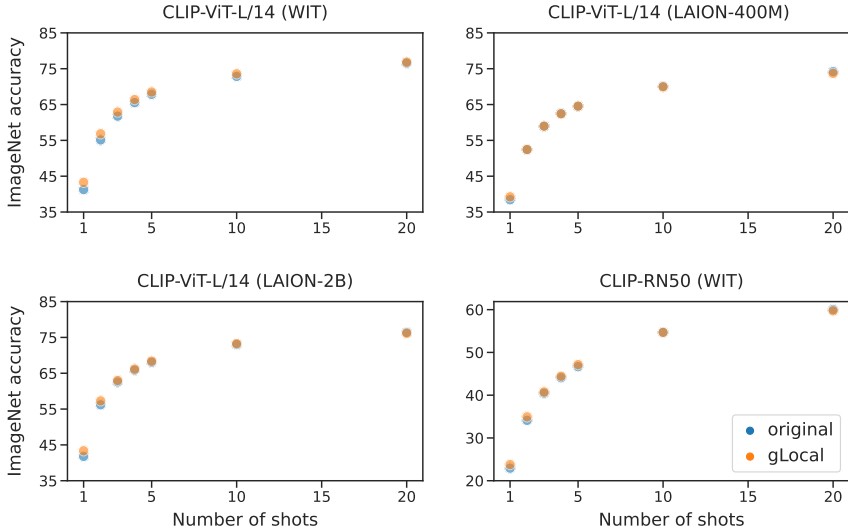

Figure 12: ImageNet validation set accuracy as a function of the number of training examples per class used for training a linear classifier on the original and the gLocal-transformed representations. For all CLIP models, we report the mean validating set accuracy across three different seeds.

## D.2 Anomaly detection

In addition to the results of image/text models for the "one-vs-rest" anomaly detection (AD) setting that we presented in §4.4, here we show "one-vs-rest" AD performance of ImageNet models. While the gLocal transform considerably improves AD performance over the untransformed representations

across the different datasets for image/text models (see Tab. {2, 3}, for ImageNet models we do not observe any improvements over the original representation space (see Tab. {8, 9}).

Furthermore, we present results for the non-standard Leave-one-out (LOO) setting and for "CIFAR10-vs-CIFAR100" for all image/text and Imagenet models that we considered. In the "CIFAR10-vs-CIFAR100" AD task, all data of CIFAR10 is considered to be the normal class, and each sample from the CIFAR100 dataset is considered an anomaly. Similarly to the previously reported AD results, the gLocal transform substantially improves AD performance compared to the original representations for image/text models across all datasets but does not appear to have a considerable impact on the performance of ImageNet models (see Tab. {10, 11}.

Table 8: One-vs-rest nearest neighbor based AD results; with and without transformation. ImageNet30 results for ImageNet models are omitted due to overlap with train data.

| Model \ Transform | CIFAR10 | | CIFAR100 | | CIFAR100-Coarse | | ImageNet30 | | DTD | |
|---|---|---|---|---|---|---|---|---|---|---|
| | original | gLocal | original | gLocal | original | gLocal | original | gLocal | original | gLocal |
| AlexNet | **89.43** | 85.63 | **92.34** | 88.53 | **87.53** | 82.75 | - | - | **86.33** | 79.51 |
| ResNet-18 | **92.19** | 84.96 | **95.06** | 89.71 | **92.16** | 84.89 | - | - | **94.38** | 89.55 |
| ResNet-50 | **94.74** | 89.74 | **96.46** | 93.76 | **94.3** | 91.2 | - | - | **94.47** | 91.52 |
| VGG-16 | **90.33** | 88.0 | **93.56** | 91.97 | **89.78** | 88.16 | - | - | **91.15** | 85.5 |

Table 9: One-vs-rest AD with a class distribution shift between train and test sets; with and without transformation.

| Model \ Transform | Entity-13 | | Entity-30 | | Living-17 | | Nonliving-26 | | Cifar100-shift | |
|---|---|---|---|---|---|---|---|---|---|---|
| | original | gLocal | original | gLocal | original | gLocal | original | gLocal | original | gLocal |
| AlexNet | **83.84** | 81.45 | **85.38** | 83.71 | **87.04** | 79.09 | **81.45** | 78.84 | **80.21** | 76.37 |
| ResNet-18 | **91.84** | 88.01 | **93.18** | 90.32 | **96.82** | 89.11 | **90.97** | 88.82 | **81.83** | 76.84 |
| ResNet-50 | **89.59** | 87.24 | **93.51** | 89.63 | **98.27** | 89.87 | **90.61** | 88.65 | **84.73** | 84.45 |
| VGG-16 | **89.78** | 88.87 | 90.7 | **91.56** | **94.72** | 89.98 | **89.78** | 89.32 | **83.42** | 81.91 |

Table 10: LOO nearest neighbor based AD results and "CIFAR-10 vs. CIFAR-100" AD results; with and without using the gLocal transform.

| Model \ Transform | CIFAR10 | | CIFAR100 | | Cifar100-Coarse | | Cifar10 vs Cifar100 | |
|---|---|---|---|---|---|---|---|---|
| | original | gLocal | original | gLocal | original | gLocal | original | gLocal |
| AlexNet | **67.64** | 62.7 | **58.94** | 55.83 | **63.33** | 59.37 | **69.87** | 68.01 |
| ResNet-18 | **72.35** | 63.47 | **64.86** | 58.53 | **71.38** | 62.89 | **81.42** | 73.71 |
| ResNet-50 | **76.62** | 68.27 | **66.91** | 61.26 | **74.78** | 68.63 | **84.27** | 80.17 |
| VGG-16 | **68.45** | 64.31 | **59.92** | 57.89 | **65.81** | 64.28 | 73.55 | **74.7** |
| CLIP-RN50 | 70.32 | **72.46** | 59.91 | **61.43** | 65.63 | **68.07** | 72.55 | **76.78** |
| CLIP-ViT-L/14 (WIT) | 84.91 | **91.3** | 67.08 | **72.24** | 73.48 | **80.7** | 85.24 | **92.78** |
| CLIP-ViT-L/14 (LAION-400M) | **93.0** | 92.37 | 74.05 | **74.15** | 82.13 | **82.88** | 94.44 | **94.68** |
| CLIP-ViT-L/14 (LAION-2B) | 93.55 | **95.23** | 76.88 | **77.46** | 84.67 | **85.78** | 93.18 | **95.26** |

Table 11: LOO nearest neighbor based AD results; with and without using the gLocal transform.

| Model \ Transform | Entity-13 | | Entity-30 | | Living-17 | | Non-Living-26 | |
|---|---|---|---|---|---|---|---|---|
| | original | gLocal | original | gLocal | original | gLocal | original | gLocal |
| AlexNet | **62.05** | 59.73 | **58.2** | 56.23 | **61.02** | 56.07 | **56.27** | 54.93 |
| ResNet-18 | **74.26** | 68.88 | **70.6** | 64.7 | **76.48** | 70.79 | **66.61** | 63.82 |
| ResNet-50 | 72.46 | **73.67** | 70.6 | 72.49 | **83.5** | 82.45 | 68.16 | **68.29** |
| VGG-16 | **70.26** | 68.03 | **66.38** | 63.24 | **73.31** | 65.99 | **64.87** | 62.95 |
| CLIP-RN50 (WIT) | 69.99 | **71.12** | 63.49 | **63.72** | 72.52 | 70.04 | 62.55 | **63.4** |
| CLIP-ViT-L/14 (WIT) | 71.68 | **74.88** | 68.96 | **70.58** | **82.9** | 82.72 | 63.94 | **67.33** |
| CLIP-ViT-L/14 (LAION-400M) | 69.98 | **71.44** | 65.68 | **66.26** | 77.35 | **77.4** | 65.19 | **66.27** |
| CLIP-ViT-L/14 (LAION-2B) | 70.77 | **72.49** | 66.55 | **67.68** | 80.07 | **80.09** | 65.43 | **67.99** |

**The nearest neighbor hyperparameter** $k$. From the results reported in Tab. 12 it can be inferred that the nearest neighbor hyperparameter $k$ does not have a considerable impact on AD task performance across the different datasets. Here, we report the impact of $k$ on the performance of CLIP ViT-L/14 (WIT) but the observation holds across all image/text models.

Table 12: Nearest Neighbor AD performance of CLIP ViT-L/14 for different $k$.

| $k$ | 2 | | 5 | | 10 | | 20 | |
|---|---|---|---|---|---|---|---|---|
| Dataset \ Transform | original | gLocal | original | gLocal | original | gLocal | original | gLocal |
| CIFAR-10 | 95.37 | **98.16** | 95.14 | **98.16** | 94.86 | **98.11** | 94.50 | **98.03** |
| CIFAR-100 | 91.90 | **97.22** | 91.41 | **97.19** | 90.93 | **97.08** | 90.39 | **96.92** |
| CIFAR-100-coarse | 89.28 | **95.9** | 88.50 | **95.83** | 87.73 | **95.66** | 86.81 | **95.41** |
| CIFAR-100-shift | 74.48 | **87.06** | 73.69 | **87.11** | 73.00 | **87.01** | 72.29 | **86.85** |
| ImageNet30 | 98.95 | **99.74** | 98.91 | **99.75** | 98.85 | **99.77** | 98.78 | **99.77** |
| Entity-13 | 88.37 | **92.5** | 88.54 | **93.23** | 88.45 | **93.63** | 88.28 | **93.95** |
| Entity-30 | 91.26 | **95.11** | 91.31 | **95.53** | 91.22 | **95.74** | 91.03 | **95.91** |

## D.3 Global versus gLocal transform

Aside from the AD performance of CLIP RN50 and CLIP ViT-L/14 (WIT), the gLocal transform leads to more substantial improvements on downstream tasks than the global transform. In Tab 13, we report the average few-shot and anomaly detection performances using the global or gLocal transforms. For FS, we average performance over all results reported in Tab. 1, and for AD we average performance across all results reported in Tab. {2, 3, 6}.

Table 13: Comparison of the average downstream task performance global and gLocal transforms.

| | AD | | FS | |
|---|---|---|---|---|
| Model \ Transform | global | gLocal | global | gLocal |
| AlexNet | 81.16 | **81.76** | 28.43 | **28.91** |
| ResNet-18 | 84.62 | **86.91** | 43.42 | **44.61** |
| ResNet-50 | **93.19** | 89.56 | **51.32** | 50.55 |
| VGG-16 | 87.32 | **88.36** | 37.23 | **39.25** |
| CLIP-RN50 (WIT) | **92.12** | 91.52 | 54.63 | **55.00** |
| CLIP-ViT-L/14 (WIT) | **95.49** | 95.19 | **69.92** | 69.91 |
| CLIP-ViT-L/14 (LAION-400M) | 95.72 | **96.08** | 68.80 | **69.40** |
| CLIP-ViT-L/14 (LAION-2B) | 96.33 | **96.65** | 72.00 | **72.27** |

## D.4 Human-adversarial experiments

To rule out confounding variables other than global similarity structure for improved downstream task performance of the gLocal transform, we created a new triplet dataset where for each triplet in the data we choose an object that is different from the original human choice to be the new odd-one-out choice. Note that this is not a random choice over all objects but a random choice over the set of two objects not chosen by a human participant, i.e., human-adversarial choices (albeit random and not deterministic). Using this human-adversarial triplet dataset, we use the same gLocal optimization as we do for the original task (see Eq. 6). We minimize a bounded cross-entropy error and determine convergence on a held-out, human-adversarial validation set as we have done for the original task. This allows us to use the same hyperparameter setting that we found is optimal for the gLocal transform (see Appx. A.2 for further details). Across the different few-shot learning and anomaly detection tasks, we find the adversarial transform performing slightly worse than the original transform and substantially worse than the gLocal transform for all CLIP models (see Fig. 13).

## E    Representational alignment

### E.1    Human similarity judgments and RSMs

**Multi-arrangement task**. Human similarity judgments for King et al. [41] and [10] were obtained by using a multi-arrangement task. In a multi-arrangement task, participants are presented with a computer screen showing images of several different objects. The participants are asked to arrange the images into semantically meaningful clusters, given the instruction that images of objects that lie close together are considered more similar. From this arrangement, one can infer pairwise (dis-)similarities of the objects and average those across all participants to obtain a representative (dis-)similarity matrix.

**Ordinal scale**. In Peterson et al. [65, 66], pairwise similarity judgments were obtained by asking human participants to rate the similarity of pairs of objects on an ordinal scale that ranges from 0 ("not similar at all") to 10 ("very similar"). The pairwise similarity ratings can be averaged across the different participants which in turn yields a matrix of similarities between pairs of objects.

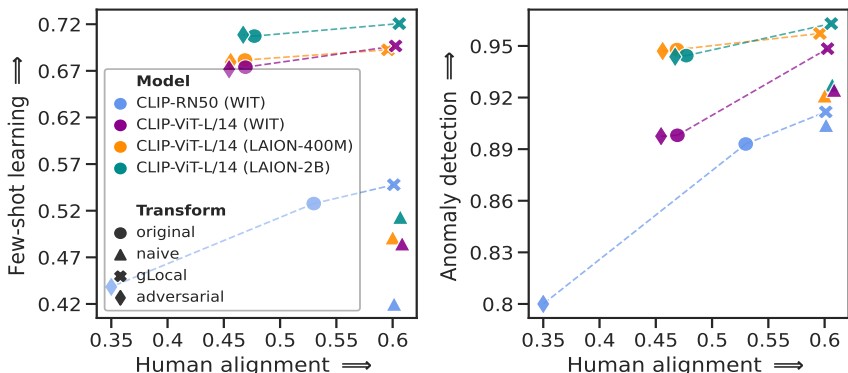

Figure 13: Few-shot learning and anomaly detection performances as a function of the degree of alignment with human similarity judgments for the original, naive, gLocal, and human-adversarial transforms.

**Triplet odd-one-out choices**. The triplet odd-one-out task is a commonly used task in the cognitive sciences to infer pairwise object similarity ratings [72, 31, 60]. The triplet odd-one-out task is a *three-alternative-forced-choice* task where participants are presented with three objects and have to select the one that does not fit. In contrast to the multi-arrangement task or an ordinal scale, the triplet odd-one-out task does not naturally yield a similarity matrix. A similarity matrix can be obtained, however, by learning representations for the objects being used in the task from the human responses. Variational Interpretable Concept Embeddings (VICE) — an approximate Bayesian method for inferring mental representations of object concepts from triplet odd-one-out choices — is a method that was specifically developed for that purpose. VICE uses variational inference to learn representations for the objects in the triplets by fitting the human responses via stochastic gradient descent. The method minimizes $\mathcal{L}_{\text{global}}$ with additional non-negativity and sparsity constraints on the representations. More details about the optimization can be found in Muttenthaler et al. [60]. From the VICE solution, one can easily compute a representational similarity matrix (RSM). Specifically, given learned object representations $\boldsymbol{V} \in \mathbb{R}^{n \times d}$, one first computes the dot-product similarity matrix $\boldsymbol{S}_h \coloneqq \boldsymbol{V}\boldsymbol{V}^\top$ and then exponentiate this matrix elementwise, $\boldsymbol{S}_h' \coloneqq \exp(\boldsymbol{S}_h)$. One can then apply the softmax function defined in Eq. 1 to every combination of triplets in the exponentiated similarity matrix which yields the final RSM for triplet odd-one-out choices from Hebart et al. [32]. The last step is performed to guarantee that the pairwise similarities are modeled according to the triplet odd-one-out objective function that was used to learn the human object representations $\boldsymbol{V}$ (see Eq. 2).

### E.2  Neural network representations and RSMs

**Neural network representations**. RSMs for neural network representations are obtained by first embedding the same set of images that were presented to the human participants in the $p$-dimensional latent space of a neural net. The latent space could be any layer of a neural network. Here we use the penultimate layer space for ImageNet models and the image encoder space for image/text models. We do this because previous work has shown that the penultimate layer space of ImageNet models and the image encoder space of image/text models respectively yield the highest similarity to human behavior [66, 67, 61]. After embedding the images into the neural net's latent space, one obtains a representation matrix $\boldsymbol{X} \in \mathbb{R}^{n \times p}$ for the $n$ images in the data. Instead of simply computing the dot-product similarity matrix $\boldsymbol{S} \coloneqq \boldsymbol{X}\boldsymbol{X}^\top$, in RSA one typically uses either a cosine similarity or a Pearson correlation kernel to compute the affinity matrix,

$$\cos(\boldsymbol{x}_i, \boldsymbol{x}_j) \coloneqq \frac{\boldsymbol{x}_i^\top \boldsymbol{x}_j}{||\boldsymbol{x}_i||_2 ||\boldsymbol{x}_j||_2}; \qquad \phi(\boldsymbol{x}_i, \boldsymbol{x}_j) \coloneqq \frac{(\boldsymbol{x}_i - \bar{\boldsymbol{x}}_i)^\top (\boldsymbol{x}_j - \bar{\boldsymbol{x}}_j)}{|| (\boldsymbol{x}_i - \bar{\boldsymbol{x}}_i) ||_2 || (\boldsymbol{x}_j - \bar{\boldsymbol{x}}_j) ||_2},$$

where the cosine similarity kernel function $\cos(\boldsymbol{x}_i, \boldsymbol{x}_j)$ or the Pearson correlation kernel function $\phi(\boldsymbol{x}_i, \boldsymbol{x}_j)$ is applied to every $(\boldsymbol{x}_i, \boldsymbol{x}_j)$ vector pair of the matrix $\boldsymbol{X}$ for obtaining the final representational similarity matrix $\boldsymbol{S}' \in \mathbb{R}^{n \times n}$. Here, we use the Pearson correlation kernel function $\phi(\boldsymbol{x}_i, \boldsymbol{x}_j)$ to obtain a neural net's RSM. Pearson correlation is the centered version of cosine similarity and the ranking of the obtained similarities does not differ between the two kernel functions but Pearson correlation first centers the vectors to have zero mean and is therefore a more robust measure. For obtaining RSMs with transformed representations, the transforms are first applied to $\boldsymbol{X}$ before computing $\boldsymbol{S}'$.

### E.3 Representational Similarity Analysis (RSA)

**Additional RSMs.** To corroborate our findings from §4.5, here we additionally show RSMs for CLIP RN50 and CLIP ViT-L/14 (Laion 2B). In accordance with the different RSMs obtained from the representation space of CLIP ViT-L/14 (WIT), there does not appear to be a qualitative difference in the global similarity structure between the RSMs obtained from applying either the naive or the gLocal transforms to CLIP RN50 or CLIP ViT-L/14 (Laion 2B) (see Fig. 14). Hence, the gLocal transform improves representational alignment while preserving the local similarity structure of the original representation equally well for the different CLIP models, as we show in Tab. 4.

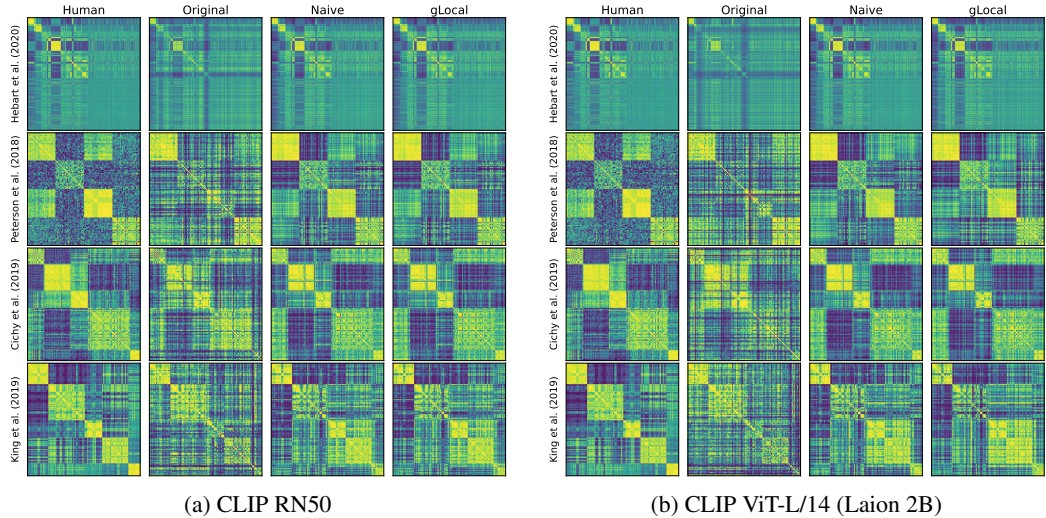

(a) CLIP RN50           (b) CLIP ViT-L/14 (Laion 2B)

Figure 14: Here, we show representational similarity matrices (RSMs) for human behavior and CLIP RN50 [WIT; 70] and CLIP ViT-L/14 [Laion 2B; 82] for four different human similarity judgment datasets [31, 66, 10, 41]. We contrast RSMs obtained from the network's original representation space (second column), the naively transformed representation space [61] (third column), and the representation space obtained by using the gLocal transform (rightmost column) against RSMs directly constructed from human similarity judgments (leftmost column).

## F  Global transform derivation

Here we derive that

$$\min_{\alpha} \|\boldsymbol{W} - \alpha I\|_{\mathrm{F}}^2 = \left\|\boldsymbol{W} - \left(\sum_{i=1}^{p} \boldsymbol{W}_{ii}/p\right) I\right\|_{\mathrm{F}}^2 .$$

First, observe that

$$
\begin{aligned}
\min_{\alpha} \|\boldsymbol{W} - \alpha I\|_{\mathrm{F}}^2 &= \min_{\alpha} \sum_{i=1}^{p} \sum_{j=1}^{p} \left(\boldsymbol{W}_{ij} - \alpha \mathbb{1}_{[i=j]}\right)^2 \\
&= \min_{\alpha} \sum_{i=1}^{p} \sum_{j=1, j\neq i}^{p} \boldsymbol{W}_{ij}^2 + \sum_{k=1}^{p} \left(\boldsymbol{W}_{kk} - \alpha\right)^2 \\
&= \sum_{i=1}^{p} \sum_{j=1, j\neq i}^{p} \boldsymbol{W}_{ij}^2 + \min_{\alpha} \sum_{k=1}^{p} \left(\boldsymbol{W}_{kk} - \alpha\right)^2 .
\end{aligned}
$$

The minimizer of $\min_\alpha \sum_{k=1}^p (\boldsymbol{W}_{kk} - \alpha)^2$ is attained with $\alpha = \sum_{\ell=1}^p \boldsymbol{W}_{\ell\ell}/p$. Substituting this back into the last equality and reversing the steps from before we have

$$\sum_{i=1}^p \sum_{j=1, j\neq i}^p \boldsymbol{W}_{ij}^2 + \min_\alpha \sum_{k=1}^p (\boldsymbol{W}_{kk} - \alpha)^2 = \sum_{i=1}^p \sum_{j=1, j\neq i}^p \boldsymbol{W}_{ij}^2 + \sum_{k=1}^p \left( \boldsymbol{W}_{kk} - \sum_{\ell=1}^p \boldsymbol{W}_{\ell\ell}/p \right)^2$$

$$= \sum_{i=1}^p \sum_{j=1}^p \left( \boldsymbol{W}_{ij} - \left( \sum_{\ell=1}^p \boldsymbol{W}_{\ell\ell}/p \right) \mathbb{1}_{[i=j]} \right)^2$$

$$= \left\| \boldsymbol{W} - \left( \sum_{\ell=1}^p \boldsymbol{W}_{\ell\ell}/p \right) I \right\|_F^2 ,$$

which finishes our derivation.

## G  Properties of LCKA

Kornblith et al. [43] previously validated linear centered kernel alignment (LCKA) as a way to measure similarity between neural network representations. Given representations $\boldsymbol{X} \in \mathbb{R}^{n \times p}$ and $\boldsymbol{Y} \in \mathbb{R}^{n \times p_2}$ containing embeddings of the same $n$ images stacked row-wise, LCKA is:

$$\text{LCKA}(\boldsymbol{X}, \boldsymbol{Y}) = \frac{\langle \tilde{\boldsymbol{X}}\tilde{\boldsymbol{X}}^\top, \tilde{\boldsymbol{Y}}\tilde{\boldsymbol{Y}}^\top \rangle_F}{\|\tilde{\boldsymbol{X}}\tilde{\boldsymbol{X}}^\top\|_F \|\tilde{\boldsymbol{Y}}\tilde{\boldsymbol{Y}}^\top\|_F} = \frac{\|\tilde{\boldsymbol{X}}^\top \tilde{\boldsymbol{Y}}\|_F^2}{\|\tilde{\boldsymbol{X}}^\top \tilde{\boldsymbol{X}}\|_F \|\tilde{\boldsymbol{Y}}^\top \tilde{\boldsymbol{Y}}\|_F}, \tag{7}$$

where $\tilde{\boldsymbol{X}}$ and $\tilde{\boldsymbol{Y}}$ are equal to $\boldsymbol{X}$ and $\boldsymbol{Y}$ with column means subtracted. (Formally, $\tilde{\boldsymbol{X}} = \boldsymbol{H}\boldsymbol{X}$ and $\tilde{\boldsymbol{Y}} = \boldsymbol{H}\boldsymbol{Y}$ and $\boldsymbol{H} = \boldsymbol{I} - \frac{1}{n}\boldsymbol{1}\boldsymbol{1}^\top$ is the centering matrix, which is a matrix representation of the linear operator that subtracts column means.)

As Kornblith et al. [43] note, linear CKA can be thought of as measuring the cosine similarity between all pairs of principal components (PCs) of $\tilde{\boldsymbol{X}}$ and $\tilde{\boldsymbol{Y}}$, weighted by the products of the proportions of variance these PCs explain in each representation. Formally, let $\tilde{\boldsymbol{X}} = \boldsymbol{U}\boldsymbol{\Sigma}\boldsymbol{V}^\top$ and $\tilde{\boldsymbol{Y}} = \boldsymbol{U}'\boldsymbol{\Sigma}'\boldsymbol{V}'^\top$ be the singular value decompositions of $\tilde{\boldsymbol{X}}$ and $\tilde{\boldsymbol{Y}}$. The left-singular vectors $\boldsymbol{u}_i = \boldsymbol{U}_{:,i}$ are the (unit-norm) PCs of $\boldsymbol{X}$, and the squared singular values $\lambda_i = \Sigma_{ii}^2$ are the amount of variance that those PCs explain (up to a factor of $1/n$). Given these singular value decompositions, linear CKA is:

$$\text{LCKA}(\boldsymbol{X}, \boldsymbol{Y}) = \frac{\sum_{i=1}^{p_1} \sum_{j=1}^{p_2} \lambda_i \lambda_j' \left( \boldsymbol{u}_i^\top \boldsymbol{u}_j' \right)^2}{\sqrt{\sum_{i=1}^{p_1} \lambda_i^2} \sqrt{\sum_{j=1}^{p_2} \lambda_j'^2}}. \tag{8}$$

