original | CIFAR10 gLocal | CIFAR100 original | CIFAR100 gLocal | CIFAR100-Coarse original | CIFAR100-Coarse gLocal | ImageNet30 original | ImageNet30 gLocal | DTD original | DTD gLocal |
|---|---|---|---|---|---|---|---|---|---|---|
| AlexNet | **89.43** | 85.63 | **92.34** | 88.53 | **87.53** | 82.75 | _ | _ | **86.33** | 79.51 |
| ResNet-18 | **92.19** | 86.70 | **95.06** | 90.89 | **92.16** | 86.38 | _ | _ | **94.38** | 90.11 |
| ResNet-50 | **94.74** | 94.13 | **96.46** | 96.18 | **94.3** | 94.03 | _ | _ | **94.47** | 94.42 |
| VGG-16 | **90.33** | 88.00 | **93.56** | 91.97 | **89.78** | 88.16 | _ | _ | **91.15** | 85.5 |

Table D.3: One-vs-rest AD with a class distribution shift between train and test sets; with and without transformation.

| Model \ Transform | Entity-13 original | Entity-13 gLocal | Entity-30 original | Entity-30 gLocal | Living-17 original | Living-17 gLocal | Nonliving-26 original | Nonliving-26 gLocal | Cifar100-shift original | Cifar100-shift gLocal |
|---|---|---|---|---|---|---|---|---|---|---|
| AlexNet | **83.84** | 81.45 | **85.38** | 83.71 | **87.04** | 79.09 | **81.45** | 78.84 | **80.21** | 76.37 |
| ResNet-18 | **91.84** | 89.45 | **93.18** | 91.6 | **96.82** | 93.1 | **90.97** | 89.87 | **81.83** | 77.44 |
| ResNet-50 | 89.59 | **91.26** | 93.51 | **93.86** | **98.27** | 97.98 | 90.61 | **91.85** | 84.73 | **85.38** |
| VGG-16 | **89.78** | 88.87 | 90.7 | **91.56** | **94.72** | 89.98 | **89.78** | 89.32 | **83.42** | 81.91 |