# OpenReview forum: "Improving neural network representations using human similarity judgments"
_NeurIPS.cc/2023/Conference — NeurIPS 2023 poster_

### Official Review · Reviewer_BF6t · 2023-07-03

**Soundness:** 4 excellent
**Presentation:** 4 excellent
**Contribution:** 3 good
**Rating:** 7
**Confidence:** 3

**Summary:**

The paper proposes a new objective function on the representation level that enforces a global structure similar to that of humans, while maintaining as much as possible the original local structure. The goal is to maintain a separability of classes (ie keep local structure) while placing similar classes close to each other (ie enforce a global structure similar to humans), eg, cats of different breads should be placed more closely to each other than raccoons (global), while they should still be separable (local).
Three main losses are proposed: (1) naive transform, which only maximizes the alignment between a NNs representation and human similarity judgements, ie only enforces a new human-like global structure; (2) global transform, which regularizes the transformed representation toward the original one,  ; (3) gLocal transform, which minimizes a combination of a global alignment loss while also maintaining a local structure with the use of a local contrastive loss.
The authors show that gLocal transform preserves the local structure of the original representation space while capturing the same global structure as the naive transform. They demonstrate that the gLocal transform improves performance on few-shot learning and anomaly detection tasks. The naive transform approach on the other hand does not improve performance. Finally, they compare the alignment of the transformed representations with human similarity judgment datasets and find that the gLocal transform yields only slightly worse alignment than the naive transform.

**Strengths:**

The paper is well-written, clear and has an easy-to-follow flow. The method also has originality in the way that the authors piece the components of the losses together. The experiments sound

**Weaknesses:**

The results only have marginal improvements over baseline.

**Questions:**

* Why is LCKA the chosen RSM? The motivation behind the use of this measure could be made more clear in the text.
* Why was RSM chosen for the final analysis on alignment with human judgments? There are many other representation similarity measures, e.g., CKA [1].
* Why not test the representations on transfer learning (not only few shot learning or anomaly detection)?
* Is there a way to quantify the t-SNE and PCA plots (e.g., Pearson correlation between them)? Sometimes only looking at images might lead to misleading conclusions.

[1] Kornblith, Simon, et al. "Similarity of neural network representations revisited." International conference on machine learning. PMLR, 2019.

**Limitations:**

The authors have adequately addressed the clear limitations of the proposed method.

---

> ### Author Rebuttal · Authors · 2023-08-09
>
> **Responses**
>
> > Why is LCKA the chosen RSM? The motivation behind the use of this measure could be made more clear in the text
>
> The appropriate representational similarity measure here ultimately relates to how one defines “global representational structure” mathematically. If one defines global representational structure as the dot products or Euclidean distances (since $2x^\top y = ||x||^2 + ||y||^2 - ||x - y||^2$) between the representations of the individual examples, then LCKA is a natural way to measure the similarity of global representational structure. The RBF kernel and other kernel functions that decay to 0 with distance are not suitable because our representational similarity measure should care about whether distances between points match even when they are far. Regression and CCA-based similarity measures are not suitable as they rescale directions in one or both representation spaces, and after doing so, points that were far apart could be close together. Although excluding these families of measures still leaves many to choose from, LCKA is arguably the simplest. Furthermore, the decomposition of LCKA in Appendix G seems intuitively appropriate to our case: LCKA effectively measures the alignment between pairs of PCs, putting greater weight on PCs that explain more variance.
>
> > Why was RSM chosen for the final analysis on alignment w/ human judgments? There are many other representation similarity measures, e.g., CKA
>
> RSA is a commonly used technique for comparing neural network representations (or essentially any latent representation) with human similarity judgments. In a multi-arrangement task, human participants are asked to freely arrange objects on a computer screen according to their *perceived* similarity [1,2]. These pairwise distances are subsequently aggregated across the different participants.
>
> CKA is a specific flavor of RSA where the RSMs/kernel matrices are positive semi-definite (PSD), the means are subtracted from the rows and columns, and the similarity between the RSMs is measured using cosine similarity (see Appendices E and G in the Supplementary Material for details on RSA and CKA respectively). The aggregated pairwise distances from the multi-arrangement task cannot necessarily be converted to a PSD RSM; a relationship between distance matrices and PSD similarity matrices exists only when the distance is a semi-metric of negative type [3]. The human pairwise similarity data obtained by Peterson et al. (2016; 2018) likewise does not result in a PSD matrix. Rather than try to justify the use of non-PSD matrices with CKA, we decided to use the same flavor of RSA as the work that introduced these datasets. This amounts to constructing a similarity matrix for the neural net representations using Pearson correlation between embeddings and measuring similarity between the RSMs using Pearson correlation. Based on our past experiences, there is little qualitative difference between this flavor of RSA and CKA.
>
> > Why not test the representations on transfer learning (not only few shot learning or anomaly detection)?
>
>
> FS and AD are commonly used downstream tasks that evaluate the transferability of pretrained representations (see response to Reviewer 2yQv). In addition, we've performed linear probing on ImageNet (see PDF in general response).
>
> > Is there a way to quantify the t-SNE and PCA plots (e.g., Pearson correlation between them)? Sometimes only looking at images might lead to misleading conclusions.
>
> Although we could’ve stated this more explicitly, in Figures 2 and 3 of the submission, we directly quantify the underlying high-dimensional structure that the t-SNE and PCA plots attempt to represent in low-dimensional space.
>
> t-SNE attempts to reduce the dimensionality of the data such that neighborhood relationships are preserved. In the first paragraph of Section 4.2 and Figure 2, we directly quantify the extent to which neighborhood relationships are preserved, observing much better preservation for the gLocal/global probes, in line with the t-SNE plots.
>
> If we took the dot products between all of the pairs of points in each PCA plot and measured the cosine similarity between those dot products, the result would correspond to CKA after setting the singular values of each representation to zero for all but the two largest PCs. We perform a similar quantification in Figure 3 using the 10 largest PC. Using the top 2 PCs yields qualitatively similar results to using the top 10 PCs.
>
> For completeness, we have also measured the mean-squared Euclidean distances between the points in the plots after standardization to unit variance in the table below. The conclusions match both those in Figures 2 and 3 and the qualitative appearance of the plots: the gLocal PCA plot closely resembles the naive PCA plot, whereas the gLocal t-SNE plot closely resembles the untransformed t-SNE plot.
>
> |                        | PCA           | t-SNE         |
> |------------------------|---------------|---------------|
> | Untransformed vs. gLocal |  0.14  | 0.05 |
> | Untransformed vs. Naive  | 0.19 | 0.20 |
> | gLocal vs. Naive         | 0.02 | 0.18 |
>
>
> **References**
>
> [1] Radoslaw M. Cichy, Nikolaus Kriegeskorte, Kamila M. Jozwik, Jasper J.F. van den Bosch, and Ian Charest. The spatiotemporal neural dynamics underlying perceived similarity for real-world objects. *NeuroImage*, 194:12–24, 2019. ISSN 1053-8119. doi:
> https://doi.org/10.1016/j.neuroimage.2019.03.031.
>
> [2]  Marcie L. King, Iris I.A. Groen, Adam Steel, Dwight J. Kravitz, and Chris I. Baker. Similarity judgments and cortical visual responses reflect different properties of object and scene categories in naturalistic images. *NeuroImage*, 197:368–382, 2019. ISSN 1053-8119. doi: https://doi.org/ 493 10.1016/j.neuroimage.2019.04.079.
>
> [3] Sejdinovic, D., Sriperumbudur, B., Gretton, A., & Fukumizu, K. (2013). Equivalence of distance-based and RKHS-based statistics in hypothesis testing. *The Annals of Statistics*, 2263-2291.

---

> > ### Author Response · Authors · 2023-08-19
> >
> > Dear reviewer BF6t,
> >
> > is there anything else we can clarify before the discussion period ends on August 21st?
> >
> > Best
> >
> > Submission5272 Authors

---

> > > ### Comment · Reviewer_BF6t · 2023-08-19
> > >
> > > I thank the authors for their rebuttal. My concerns have been addressed and all details have been clarified. I maintain my score.

---

### Official Review · Reviewer_dfyY · 2023-07-05

**Soundness:** 2 fair
**Presentation:** 3 good
**Contribution:** 3 good
**Rating:** 6
**Confidence:** 4

**Summary:**

This paper proposes a linear transformation named gLocal transform to align the global structure of the representation space with human similarity judgments, while at the same time maintains the local structure of the original representation space. The method is shown to be superior than a naive transform that minimizes only the global alignment loss and increases performance on few-shot learning and anomaly detection tasks.

**Strengths:**

It is interesting to investigate the global structure of representation space and the correlation with human judgements. The paper further shows its practical value in down-stream tasks. The idea is novel. The experimental design is also interesting that verifies the human alignment of gLocal and naively transformed representations using four human similarity judgment datasets.

**Weaknesses:**

In regards to the experiments on few-shot learning, it would be beneficial to not only compare the results with CLIP experiments but also include experiments with general baselines. This will provide a comprehensive evaluation and determine if the proposed approach consistently improves the results compared to existing methods.



**Questions:**

See Weaknesses above.

**Limitations:**

The paper has discussed the limitations.

---

> ### Author Rebuttal · Authors · 2023-08-08
>
> **Response**
>
>
> > In regards to the experiments on few-shot learning, it would be beneficial to not only compare the results with CLIP experiments but also include experiments with general baselines. This will provide a comprehensive evaluation and determine if the proposed approach consistently improves the results compared to existing methods.
>
>
> We refer the reviewer to Section D in the Supplementary Material, where we provide numerous experimental results for general baselines such as supervised models trained on ImageNet (e.g., ResNet50, VGG16, Alexnet). We find that the gLocal transform does not work as well for ImageNet models as it does for CLIP models. The gLocal transform does not seem to find a *best-of-both-worlds* representation space for ImageNet models. We don’t think that this is too surprising given that the visual information encoded in the representations of CLIP models is much richer (due to dataset diversity + multi-model pretraining) compared to the representations of ImageNet models (fewer data points + unimodal pretraining). Therefore, global similarity structure can be easier decoded from the CLIP representations vs. ImageNet model representations, while at the same time preserving local similarity structure. We explain our findings in the Discussion section of the main text. See also Muttenthaler et al. (2023) [1] for evidence on the importance of training data + objective function (vs. architecture + model size) for alignment with human similarity judgments.
>
> [1] Lukas Muttenthaler, Jonas Dippel, Lorenz Linhardt, Robert A Vandermeulen, and Simon Kornblith. Human alignment of neural network representations. In *11th International Conference on Learning Representations, ICLR 2023, Kigali, Rwanda, Mai 01-05, 2023*. OpenReview.net, 2023.

---

> > ### Comment · Reviewer_dfyY · 2023-08-14
> >
> > Thanks for the response. No more concerns.

---

### Official Review · Reviewer_eWR8 · 2023-07-05

**Soundness:** 3 good
**Presentation:** 3 good
**Contribution:** 3 good
**Rating:** 6
**Confidence:** 4

**Summary:**

This paper proposes to learn a linear transformation on top of pre-trained CLIP representations to improve alignment with human odd-one-out judgments from the things dataset. To do this, a similarity matrix is constructed based on pairwise similarities between the three images, and the representation is optimized so that the most similar pair matches the human decision. The proposed gLocal transformation combines the global loss, which encourages alignment with human annotations, with a local loss, which encourages the transformed representations to be similar to the untransformed representations. Evaluations are performed on downstream few-shot and anomaly detection tasks.

**Strengths:**

- The idea is clear and intuitively makes sense overall.
- There are extensive experiments that evaluate a number of datasets for few-shot and anomaly detection benchmarks, and various model backbones.
- The learned representations are analyzed using PCA, t-SNE, and CKA visualization. I found these analyses interesting to understand the changes that the representation undergoes during the training process.
- Alignment to human preferences is demonstrated across multiple human annotated datasets.


**Weaknesses:**

- The results seem to vary by dataset and by model - the benefits are mostly on the CLIP model, and ImageNet backbones are less consistent in their benefits when using the gLocal objective. I would be curious if similar trends would hold for OpenCLIP or DINO backbones.
- In most cases, the gLocal objective is compared against the naive objective (for example, Fig 1b, table 4). It would be helpful to also compare to the global objective (Eqn 4), or perhaps have a separate table that evaluates each component of the final loss function on a fixed set of tasks.

**Questions:**

- I'm not clear on why the second part of table 4 is included with comparisons on other human similarity datasets, and not with the respective tables 1 and 2? It seemed to be an average of the different tasks presented in the earlier tables.
- The experiments in the paper focus mostly on few-shot and anomaly detection settings. How does using the gLocal transformation perform on ImageNet linear probing or other transfer learning tasks?


**Limitations:**

Limitations are adequately addressed. The collected images span a subset of human similarity dimensions, and these may not necessarily align with ImageNet representations.

---

> ### Author Rebuttal · Authors · 2023-08-08
>
> **Responses**
>
>
> > The results seem to vary by dataset and by model - the benefits are mostly on the CLIP model, and ImageNet backbones are less consistent in their benefits when using the gLocal objective. I would be curious if similar trends would hold for OpenCLIP or DINO backbones.
>
> We report results for OpenCLIP models in our submission; these are the LAION-400M and LAION-2B models in our tables. The other two CLIP models are CLIP models that are trained on WIT. Since all of them are trained using the same objective function (Contrastive Language-Image Pretraining), they are generally referred to as CLIP models. To emphasize the difference between the two sets of models we report the dataset that they were trained on in brackets (WIT vs. LAION-X). Note that the dataset is the main difference between CLIP and OpenCLIP models.
>
> The results do not vary much across the different CLIP $\cup$ OpenCLIP models (see Figure 1 and Table 4) nor do they vary among ImageNet models (see tables in Appendix D). However, they vary between CLIP $\cup$ OpenCLIP and ImageNet models. While we observed that achieving a *best-of-both-worlds* representation space — i.e., improved human alignment + better downstream task performance — generally works well for all CLIP $\cup$ OpenCLIP models that we considered (see Figure 1), we were not able to obtain such a representation space for ImageNet models (see Appendix D in the Supplementary Material).
>
> We hypothesize that this is because the visual information encoded in the representations of CLIP $\cup$ OpenCLIP is much richer (due to dataset diversity + multi-model pretraining) compared to the representations of ImageNet models (fewer data points + unimodal training). Therefore, global similarity structure can be easily decoded from the CLIP representations, and at the same time we can preserve local similarity structure. There’s recent research that has shown that objective function and pretraining dataset have a much larger impact on the degree of alignment with human perception than the specific architecture or the size (width + depth) of a model [1] which may in part explain the differences that we observe between CLIP $\cup$ OpenCLIP and ImageNet models. We provide an explanation for our findings in the Discussion section.
>
>
>
> > In most cases, the gLocal objective is compared against the naive objective (for example, Fig 1b, table 4). It would be helpful to also compare to the global objective (Eqn 4), or perhaps have a separate table that evaluates each component of the final loss function on a fixed set of tasks.
>
> We performed this ablation. We report results in the Supplementary Material in A2 and D3 (please see Figure A.1 and Table D.7).
>
> > I'm not clear on why the second part of table 4 is included with comparisons on other human similarity datasets, and not with the respective tables 1 and 2? It seemed to be an average of the different tasks presented in the earlier tables.
>
> We include the averages to make it easy to contrast human similarity with the downstream task accuracy for each transform type and to be in line with the results reported in Figure 1. We'll add a note that states this explicitly to the figure caption. It’s mainly done for better readability. In addition, we will report the naive transform accuracy for each task separately in the Supplementary Material.
>
> > The experiments in the paper focus mostly on few-shot and anomaly detection settings. How does using the gLocal transformation perform on ImageNet linear probing or other transfer learning tasks?
>
> We’ve looked into this and just performed linear probing on ImageNet where we varied the number of samples per class for training the linear probe. For few-shot probing ($k<10$) on ImageNet we observe that the gLocal transforms generally outperform the original representations by a substantial margin, whereas for larger $k$ ($k>10$), we find that the gLocal transform and the original representations perform similarly well. This is not too surprising given that, since a linear transformation of the representation changes the linear probe solution by changing the impact of (explicit and implicit) regularization. Regularization matters most in low data regimes and becomes less relevant as the number of data points increases. Taking $\lim_{k \to \infty}$, the optimal model has no regularization and a linear transformation has no impact. Please see Figure 2 in the PDF attached to the general response for a plot that shows ImageNet linear probing performance as a function of different $k$, where $k$ refers to the number of samples per class used for training the probe. We’ve also evaluated the gLocal transform using the entire ImageNet training set and observed that it performs as well as the original CLIP representations for the full data setting (see the table at the bottom of the PDF document attached to the general response).
>
>
> **References**
>
> [1] Lukas Muttenthaler, Jonas Dippel, Lorenz Linhardt, Robert A Vandermeulen, and Simon Kornblith. Human alignment of neural network representations. In *11th International Conference on Learning Representations, ICLR 2023, Kigali, Rwanda, Mai 01-05, 2023*. OpenReview.net, 2023.

---

> > ### Comment · Reviewer_eWR8 · 2023-08-11
> > **Response to author rebuttal**
> >
> > Thanks to the authors for their response. Authors have addressed my concerns and I have updated my rating accordingly.

---

### Official Review · Reviewer_2yQv · 2023-07-13

**Soundness:** 3 good
**Presentation:** 3 good
**Contribution:** 3 good
**Rating:** 7
**Confidence:** 5

**Summary:**

This paper proposes a new alignment strategy (post-hoc) that manages to "re-align"/transform the learned feature space of modern neural networks into a new feature space that is correlated with that of humans. Authors find a way to do this by getting actual human judgment behavioural data and incorporate this in the loss function, and with only a relative low number of examples, can "correct" for the machines learned representations into a more accurate one. This paper's contributions is to show that this is: 1) actually possible with few data points ;2) show increase in performance and many computational tasks that can boost performance post-alignment correction.

**Strengths:**

* The paper is on a very important topic such as representational alignment, and very likely one of the very few works that articulate why this is important (through their empirical experiments)
* Authors provide a very clear mathematical reasoning on why their alignment procedure works (the optimization process to minimize the global distance while preserving the local distance too)
* Authors provide many experiments showing the benefit  -- across multiple datasets -- that the alignment procedure works.

**Weaknesses:**

* There is a lack of qualitative assessment, aside from the RDM visualizastion. I suggest in the question part of this review that authors run adversarial attacks to see how the perturbed stimuli look like on both the aligned and un-aligned networks.
* Somehow I feel like another control could have been used, such as an "ortho-normal" alignment e.g. preserve or maximize global distance but minimize local distance: do the results change?
* Experiments of robustness such as testing the performance of the system on style-vs-texture image fusions to test for shape bias, and/or on the ImageNet-C dataset would be really interesting. If the model is truely aligned to human perception, then the o.o.d. performance should be exceedingly higher than non-aligned networks.
*Authors focus is on accuracy. I think this is a missed opportunity to also focus more on robustness instead.

**Questions:**

* What is meant by downstream tasks?
* Is there a control for the post-hoc alignment? For example, what would have happened if you somehow maximized the global loss at alignment as a control and kept the local relations intact. Would the most still had achieved a higher performance?
* It would have been cool to do qualitative assessments of alignment. As an example: What would have happened in terms of the difference of running an adversarial attack (like PGD) on the networks pre and post alignment? Do the perturbed stimuli now fool the human? I think this would be a pretty cool finding to show such as what can be done via adversarial training as shown in Santurakar et al. (NeurIPS, 2019 with ResNets) & Berrios & Deza (BSW, 2022 with Transformers), Harrington & Deza (ICLR 2022).
* It is not obvious to me on what is the difference between the “naive” condition and “untransformed”. Both sound like the same, but I feel like I am wrong and missing something that hasn’t been really spelled out in the paper.

**Limitations:**

See questions. I overall think this is a good paper, I still do think that authors could have added more references in the NeuroAI space that do address the question of representational alignment through different training strategies (works of DiCarlo, Konkle, Krigeskorte, Yamins, etc... -- some of these works are cited, but not the most modern and relevant ones). In addition, it is worth mentioning that adversarial training in particular plays a *big* role as shown in works of Santurkar, Harrington & Deza.

I'm overall very happy/lenient to increase my score if the questions + limitations are addressed!

---

> ### Author Rebuttal · Authors · 2023-08-08
>
> **Responses**
>
> > There is a lack of qualitative assessment, aside from the RDM visualizastion.
>
> Alongside the RDM visualizations which demonstrate differences in global similarity structure between human similarity judgments and neural net representations (Figure 5 and Figure E.1), we did a thorough analysis of the local and global structure of the neural net representations for each of the conditions (see Section 4.2 *Impact of transforms on global and local structure*) where we perform PCA (*global*), t-SNE (*local*), and analyses on the nearest neighbor preservation (see Figures 2, 3, and 4). To corroborate our findings from Section 4.2, where we quantify the nearest neighbor preservation, we additionally visualize the nearest neighbor images in Appendix C in the Supplementary Material (see Figure C.1) which we think yields interesting insights into how local similarity structure is distorted by the naive transform but preserved (and sometimes even more intuitive) by the gLocal transform.
>
> Moreover, we dedicate an entire section (Appendix B in the Supplementary Material) to an analysis of the *global structure of the representations after alignment* where we investigate the movements of the representations of pairs of items from different superordinate categories from the THINGS dataset (see Figure B.1).
>
> > What is meant by downstream tasks?
>
> In a *downstream task* one typically evaluates the representations of a pretrained model on a problem different from the pretraining task. For instance, to evaluate the downstream task performance of a model that was trained on image/text pairs with a contrastive learning objective such as CLIP, one has to test the representations on a task that is different from this pretraining task.
>
> Note that this can essentially be any task one cares about (image classification, predicting fMRI signals, etc.) as long as the goal of that task is clearly defined — often with targets. In our work, the downstream tasks we cared about were all concerned with image classification or subsets thereof. Specifically, we focussed on few-shot learning and anomaly detection. We think that these are useful tasks to evaluate the representations of a pretrained model because the prior (or *initialization* of a model) matters most in low data settings.
>
> > Is there a control for the post-hoc alignment? For example, what would have happened if you somehow maximized the global loss at alignment as a control and kept the local relations intact. Would the most still had achieved a higher performance?
>
> Because adversarially training on the target task (human alignment) rather than on the auxiliary task (preserving local structure) causes problems with the optimization — there exists no lower bound on the cross-entropy error anymore because for adversarial optimization $\mathcal{L}_{\text{global}}({\bf{W}}, {\bf{b}}) \in [-\log2,-\infty)$ —,  we decided to perform a task that’s highly similar to what you’ve suggested but better for optimization: Instead of training adversarially on the original THINGS triplet dataset, we created a new triplet dataset where for each triplet in the data we choose an object that is different from the original human choice to be the new odd-one-out. Note that this is not a random choice over all objects but a random choice over the set of two objects that were not chosen by a human participant. Using this new adversarial triplet dataset, we use the same gLocal optimization as we’ve done for the original task (see Equation 6).
>
> **Observations**: For FS, we see that the adversarial transform is slightly worse than the original transform and substantially worse than the gLocal transform. For AD, the adversarial probe is worse than both the naive transform and the original representation and substantially worse than the gLocal transform.
>
> Thanks for suggesting to perform this experiment! See **Figure 1** in the PDF attached to the general response for a plot including adversarial probing results.
>
> > It is not obvious to me on what is the difference between the “naive” condition and “untransformed”. Both sound like the same, but I feel like I am wrong and missing something that hasn’t been really spelled out in the paper.
>
> As explained in the methods section of our submission (see Equation 3), the **naive transform** is a linear transformation that is learned by solely minimizing the alignment loss plus some amount of $\ell_{2}$-regularization on the transformation matrix $\bf{W}$. That is, for learning the **naive transform** we solve Equation 3. After convergence, we can use this transformation for applying it to the neural network representations before evaluating downstream task performance.
>
> In contrast to the **naive transform**, the **untransformed** (or **original** if you will) representation does not include any transformation. You can think of the transform in the *untransformed condition* as the identity matrix where $S_{ij} = (I {\bf{x}}_i)^{T} (I{\bf{x}}_j) = {\bf{x}}_i^{T}{\bf{x}}_j$. That is, we take the neural network representations as is.
>
> > [...], I still do think that authors could have added more references in the NeuroAI space [...]
>
> We agree that tying stronger connections to the NeuroAI field is useful and thus we are happy to include more recent work from the NeuroAI space (from the labs that you’ve mentioned and beyond) in the introduction and the related works section. However, since none of our experiments directly involve neuro data we didn't want to dedicate too much space to neuro literature review in the initial submission (due to space constraints), but we'll use part of the additional page in the camera ready to expand the related work section. We will also mention the works of Santurkar, Harrington & Deza which seems to be pretty interesting. Thanks for the pointer!

---

> > ### Author Response · Authors · 2023-08-19
> >
> > Dear reviewer 2yQv,
> >
> > is there anything else we can clarify before the discussion period ends on August 21st?
> >
> > Best
> >
> > Submission5272 Authors

---

### Author Rebuttal · Authors · 2023-08-09

We thank the reviewers for the overall very positive feedback. Although the reviewers’ concerns do not overlap , we would like to provide a general response where we address a few points that could be of interest to all reviewers.

**Adversarial alignment**

Reviewer 2yQv suggested to run the gLocal minimization (see Equation 6) to reduce alignment with human similarity judgments while still preserving local similarity structure in order to rule out confounding variables other than global similarity structure. We think that this is an interesting experiment and would like to thank reviewer 2yQv for their suggestion!

To perform such an analysis, we created a new triplet dataset where for each triplet in the data we choose an object that is different from the original human choice to be the new odd-one-out. Note that this is not a random choice over all objects but a random choice over the set of two objects that were not chosen by a human participant. Using this new adversarial triplet dataset, we use the same gLocal optimization as we’ve done for the original task (see Equation 6). We minimize a bounded cross-entropy error and determine convergence on a held-out validation set as we’ve done before. This allows us to use the same hyperparameter setting as we’ve used for the gLocal transform. Across the different downstream tasks, we see that the adversarial transform is slightly worse than the original transform and substantially worse than the gLocal transform for all CLIP $\cup$ OpenCLIP models. Please see Figure 1 in the attached PDF for a plot including these adversarial alignment results.


**ImageNet linear probing**

Reviewer eWR8 wanted us to evaluate the gLocal transform on ImageNet via linear probing. We performed linear probing on ImageNet where we varied the number of samples per class used for training the probe. We observe that for small sample sizes ($k<10$) the gLocal transform outperforms the original representation, but performs on par with the original representation for larger sample sizes ($k>10$). As we describe in greater detail in the response to Reviewer eWR8, given that our transformation is linear, it is expected that it becomes less relevant as the number of data points increases. Therefore most of our experiments are concerned with few-shot learning and anomaly detection (which is zero-shot). Please see Figure 2 in the attached PDF for a plot that shows ImageNet linear probing performance as a function of different $k$, where $k$ refers to the number of samples per class used during training. We also provide a table at the bottom of the PDF document (below Figure 2) including linear probing results for the ImageNet full data setting.

We address the remaining points in individual responses to the reviewers.

---

### Decision · Program_Chairs · 2023-09-21

**Decision:**

Accept (poster)

**Comment:**

The submission introduces and empirically demonstrates a framework to tune large-scale model representations via using additional human behavioral data. Positive and negative results are demonstrated and discussed, which may inspire further study. This interdisciplinary work will be of interest to several subcommunities at the conference.